# Anti-CSF-1R therapy with combined immuno-chemotherapy coordinate an adaptive immune response to eliminate macrophage enriched triple negative breast cancers

Diego A. Pedroza [1,2,3], Xueying Yuan[1,4], Fengshuo Liu [1,2,3,4], Hilda L. Chan [1,2,3,5], Christina Zhang[1], William Bowie[2,6], Alex J. Smith[1,4], Sebastian J. Calderon[1], Nadia Lieu[1], Weiguo Wu[7], Paul Porter[7], Poonam Sarkar[8], Na Zhao [1,3], Constanze V. Oehler [1], Ondrej Peller [1], M. Waleed Gaber[8], Qian Zhu [2,3,6], Charles M. Perou[9], Xiang H-F. Zhang [1,2,3,10] & Jeffrey M. Rosen [1,3] ✉

Women diagnosed with metastatic triple negative breast cancer (mTNBC) have limited treatment options, are more prone to develop resistance and are associated with high mortality. A cold tumor immune microenvironment (TIME) characterized by low T cells and high tumor associated macrophages (TAMs) in mTNBC is associated with the failure of standard-of-care chemotherapy and immune checkpoint blockade (ICB) treatment. We demonstrate that the combination of immunomodulatory low-dose Cyclophosphamide (CTX) coupled with anti-CSF-1R antibody targeted therapy (SNDX-ms6352) and anti-PD-1 (ICB), was highly effective against aggressive metastatic *Trp53* null TNBC transplantable syngeneic models that present with high macrophage infiltration. Mechanistically, CSF-1R inhibition along with CTX disrupted the M-CSF/CSF-1R axis which upregulated IL-17, IL-5 and type II interferon resulting in elevated B- and T cell infiltration. Addition of an anti-PD-1 maintenance dose helped overcome de novo PD-L1 intra-tumoral heterogeneity (ITH) associated recurrence in lung and liver mTNBC.

Triple negative breast cancer (TNBC) is the most aggressive breast cancer with an overall poorer prognosis compared to other breast cancer subtypes[1]. Newly approved immune checkpoint blockade (ICB) accompanied by cytotoxic chemotherapies can now be administered to TNBC patients[2]. While multiple clinical trials have reported positive outcomes for TNBC including those utilizing these T cell activating agents[3,4] the response rates in these patients vary and clearly more refined therapies are needed to improve patient survival especially in the metastatic setting. The Tumor Immune Microenvironment (TIME) in TNBC has been suggested to play a critical role in patient response. Tumor plasticity and the Epithelial-to-Mesenchymal Transition (EMT)[5] are critical for metastasis and a relationship exists between EMT and myeloid cells, particularly Tumor Associated Macrophages (TAMs)[6]. Results from the ARTEMIS (NCT02276443) clinical trial show that TNBC patients with a high EMT/macrophage score failed to undergo a pathological Complete Response (pCR) following neoadjuvant

chemotherapy (NACT)[7,8]. Furthermore, the clinical benefit of Immune Checkpoint Blockade (ICB) is modest and poor patient outcomes may be due in part to the presence of immunosuppressive macrophages within the TIME.

Most patients die from metastatic disease and as such metastatic breast cancer remains fatal for 97−99% of those diagnosed with a median survival of only 18−24 months[9]. In the recently published AURORA trials, profiling of paired TNBC patient samples from the primary and metastatic site have shown an immunologically quiescent TIME in sites of metastasis as compared to the primary site, with the liver having the coldest TIME. Pathway analysis done on these samples have shown a downregulation of T- and B cell receptor signaling in the metastatic sites accompanied by an abundance of macrophages and neutrophils[10,11]. Several strategies have been deployed to therapeutically manipulate TAMs within TNBCs. These include macrophage reprogramming, using Class II HDAC inhibition[12] and the LPS derivate, monophosphoryl lipid A+INFy[13], phagocytic reactivation by CD47/SIRPalpha blockade with an anti-CD47 antibody[14], and recruitment inhibition or depletion by both small molecule and antibody based targeting of either the CSF-1/CSF-1R axis[15–17] or the CCL2/CCR2 axis[18].

Previously, we reported that the combination of CTX coupled with pharmacologic inhibition of TAMs by Pexidartinib (PLX-3397), a CSF-1R small molecule inhibitor, was highly effective against aggressive, claudin-low murine mammary tumors in several syngeneic *Trp53* null TNBC models that present with high TAM infiltration. Using scRNA-seq we found that T helper cells coordinated with antigen presenting B cells to promote anti-tumor immunity and long-term responses. Using high dimensional imaging techniques, we further identified the close spatial localization of B220+ CD86+ activated B cells with CD40lg+ CD4+ T cells within tertiary lymphoid structures (TLSs) that were present up to 6 weeks post treatment in primary tumors. However, recent studies from the I-SPY 2 (NCT01042379) clinical trial demonstrated that although, the small molecule inhibitor PLX-3397 successfully depleted macrophages, off target effects on other tyrosine kinases along with liver toxicity prevented it from moving forward for the treatment of TNBC[19].

TAMs have been demonstrated to be transcriptionally distinct from their monocytic and tissue-resident macrophage counterparts[20]. Interestingly a 37-gene TAM signature associated with high CSF1 has been observed in aggressive ER/PR negative breast cancers[20,21]. Thus, inhibiting the CSF-1/CSF-1R axis remains an attractive mechanism within the TIME for the treatment of aggressive TNBCs. To overcome toxicity and off target limitations, we utilized SNDX-ms6352 (Axatilimab-csfr), a mAb with high affinity for CSF-1R, which was recently FDA approved for the treatment of chronic graft-versus-host disease (cGVHD) AGAVE-201 (NCT04710576)[22]. Using several *Trp53* null TNBC models we demonstrate that SNDX-ms6352 combined with CTX increased the levels of M-CSF along with type II interferon in primary tumors. However, ITH within the metastatic TIME of lung metastases led to recurrence with some residual tumor cells expressing increased expression of PD-L1. Thus, a maintenance dose of ICB was employed to reinvigorate the metastatic TIME of lung and liver metastases in order to prolong the overall survival and achieve a complete response (CR). Functionally, treatment cessation, tumor cell rechallenges, adaptive T cell transfer, possible TLS formation along with infiltration of tumor infiltrating lymphocytes (TILs) confirmed long-term anti-tumor adaptive immunity. These findings have provided the foundation for the combination (New) of low-dose oral cyclophoSphamide (S) To potentiate Axatilimab-csfr (A) + retifanlimab (R) in treating mTNBC (NewSTART) as a phase Ib/II clinical trial (NCT06959537).

Here, we demonstrate that targeting immunosuppressive TAMs in combination with low-dose CTX and ICB can overcome TNBC lung metastatic recurrence and promote long-term anti-tumor adaptive immunity via the accumulation of T- and B-cells of

TNBC p53 null syngeneic mouse models bearing established liver metastases.

## Results

### CSF-1R inhibition coupled with CTX leads to a type II interferon response in primary macrophage enriched TNBC pre-clinical models

To study the effects of therapeutically targeting the CSF-1/CSF-1R axis we treated several *Trp53* null TNBC syngeneic models using SNDX-ms6352, a high affinity anti-CSF-1R mAb coupled with low-dose Cyclophosphamide (CTX). These *Trp53* null syngeneic models mimic a spectrum of human breast cancer subtypes, including luminal, basal-like and claudin-low making them heterogenous[6,23]. Loss of *TP53* is observed in patients with advanced metastatic disease and is considered a biomarker of poor prognosis[24,25]. Not surprisingly up to 91% of patients diagnosed with TNBC can carry a *TP53* mutation[26].

Spatial profiling via imaging mass cytometry (IMC) revealed highly proliferative Vimentin+ tumor cells within the claudin-low T12, 2151 R, T11 models, and pan-CK+ tumor cells in the basal- 2336 R model. The claudin-low and basal-like models exhibit low CD8+ T cell infiltration and high F4/80+ TAMs, while the 2208 L luminal-like tumors had high S100A8/9+ neutrophils (Figs. 1A, B and S1A–S1C). Importantly, the claudin-low models display a mesenchymal architecture highly infiltrated by F4/80+ macrophages and express high levels of the EMT marker vimentin similar to what is observed in the patients with elevated EMT/macrophage levels.

Although, SNDX-ms6352 alone efficiently depleted F4/80+ TAMs in T12 tumors via cleaved caspase-3 activation it had little to no impact on primary tumor growth or detectable liver toxicity (Fig. S1D–S1G). Interestingly, dose-dependent accumulation of CSF-1R was observed in T12 and 2151 R TAMs in vitro (Fig. S1H and S1I), likely due to ligand-receptor binding inhibition, prohibiting internalization and downstream activation that leads to cell death. To test the long-term efficacy of SNDX-ms6352 combined with CTX we designed a dose specific treatment schedule which included treatment cessation at day 28 (Fig. 1C). TAM inhibition by SNDX-ms6352 enhanced the immunostimulatory effects of CTX resulting in increased survival of all three claudin-low (T12, 2151 R and T11) and the basal-like (2336 R) model (Fig. 1D). We observed significant complete tumor regression in the two claudin-low (T12 and 2151 R) TAM enriched models (Fig. 1D). Interestingly, only the highly neutrophil enriched luminal-like 2208 L model failed to respond to this combination treatment even though SNDX-ms6352 efficiently depleted F4/80+ TAMs in all the models (Figs. 1D and S1J). Intriguingly, both T12 and 2151 R models delayed tumor cell growth following contralateral mammary tumor cell re-challenge (Figs. 1E, F and S1K). Removal of the original and contralateral re-challenged mammary glands revealed increased infiltration of CD20+ B cells in proximity to CD8+ T cells (Fig. S1L).

Next, to capture and profile the changes within the TIME we performed whole transcriptomic single cell RNA sequencing (scRNA-seq). For unbiased analysis we utilized a barcoding technology for cell multiplexing (CellPlex), which allowed us to sequence together the two models that yielded a CR (T12 and 2151 R) (Fig. 1G). Cumulatively the combination substantially showed differences in the abundance of CD4+, CD8+ T and B cells and a decrease in macrophages (Fig. 1G, S3A and S3B). Further, we observed elevated *Mrc1* and *Vim* following CTX administration, while SNDX-ms6352 depleted, *Csf1r, Mrc1, Trem2* and *Mki67* macrophage specific transcripts (Fig. S2A). Although the combination treatment did not ablate all macrophages, only in the T12 model in this subset of macrophages was increased expression of the lipid chaperone protein *Fabp5* observed (Fig. S2B and S2D). Transcriptionally, T12 has more abundant macrophage subtypes including Prolif-TAM and lipid associated (LA)-TAMs compared to 2151 R (Fig. S2C and Supplementary datasets S1–S9). Interestingly, while CTX upregulates both PD-L1 (*CD274*) and MCH II (*H2-Ab1*) levels within the

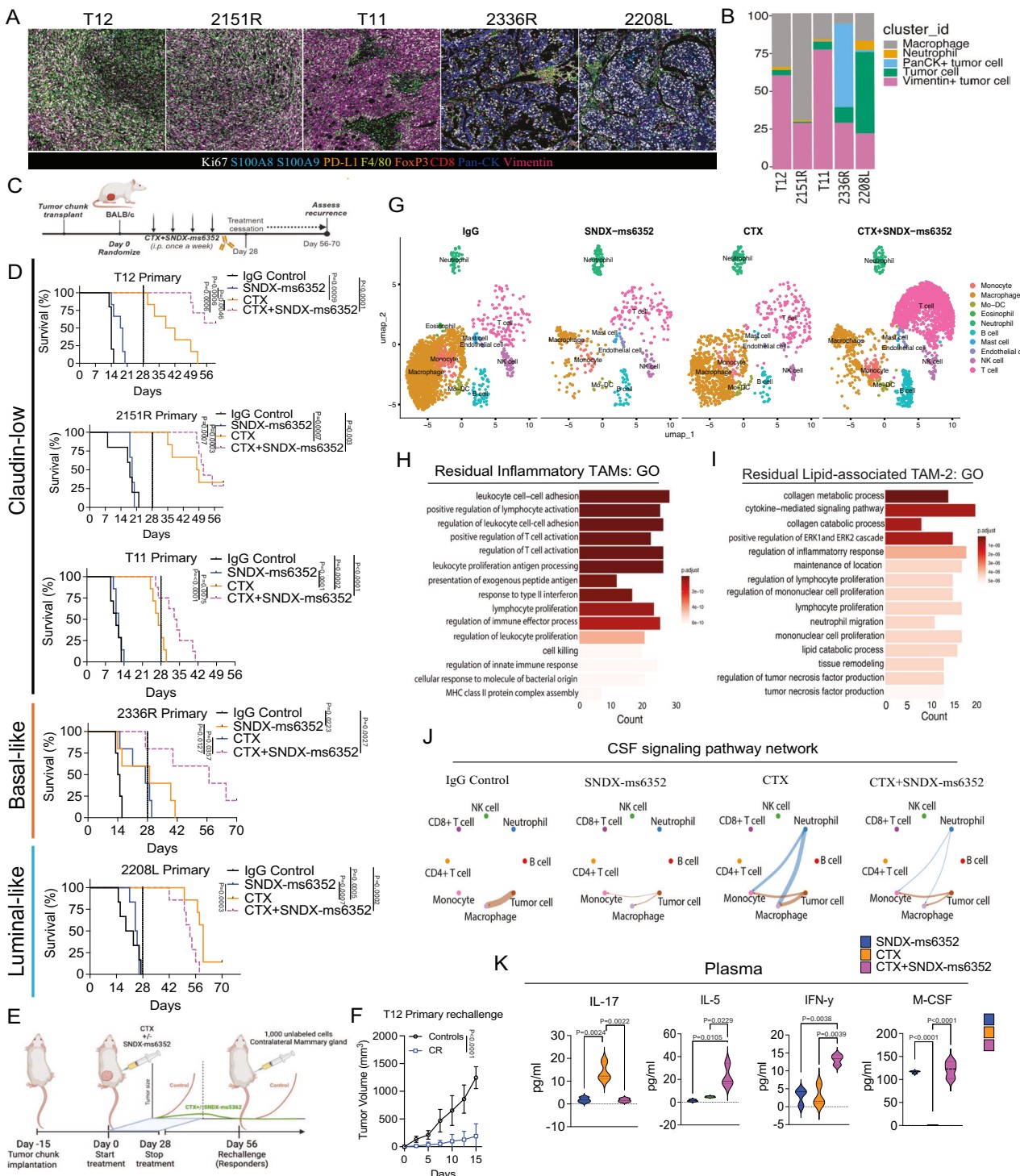

various macrophage populations (Fig. S3D, E), combination treated tumors decrease PD-L1, with residual monocytes and inflammatory-TAMs remaining MHC II positive. Gene ontology (GO) analysis of the combination treated residual inflammatory TAMs revealed increased pathway regulation and activation of T cells with interferon type II response, while lipid-associated TAMs show cytokine-mediated signaling, lymphocyte proliferation and collagen processing (Fig. 1H, I and Supplementary Dataset S10 and S11).

Only TAM depletion with either SNDX-ms6352 or the combination treatment disrupted the CSF signaling network cell-cell communication between macrophages, monocytes, neutrophils and tumor cells (Fig. 1J). Furthermore, we observed increased CCL signaling between

neutrophils and monocytes/macrophages which lead to the increase in mature macrophage populations following CTX alone (Fig. S3C). This is accompanied by increased CCL signaling between monocytes and macrophages and directly with NK cells which may lead to their suppression as previously observed (Fig. S3B).

Cytokine profiling analysis of plasma from mice bearing T12 tumors post combination treatment revealed an increase in the levels of the pro T and NK cell stimulant IL-5[27–29] type II interferon response, IFN-γ[30,31], the pro-inflammatory macrophage marker MIG (CXCL9)[32,33] and an anti-tumor response IL-4[34] (Figs. 1K, S2E and Supplementary Dataset S12). Importantly both SNDX-ms6352 and the combination treatment increased M-CSF which appears to be a functional

**Fig. 1 | CSF-1R inhibition coupled with LDMC leads to a durable response in primary macrophage enriched TNBC pre-clinical models. A** IMC analysis depicting differential expression of immune, EMT, proliferation, and exhaustion markers from different p53-/- models. Syngeneic models include the claudin-low, Basal- and luminal-like. Representative images are from a single ROI taken from a TMA. **B** Bar plots identify different cluster id's including various tumor cells within the different models. **C** Schematic representation of treatment regimen. **D** Kaplan-Meier survival curves following IgG Control, SNDX-ms6352, CTX and CTX + SNDX-ms6352. Statistical analysis performed using Log-rank (Mantel-cox) test (*n* = 5–8 mice per treatment). **E** Schematic representation of tumor cell rechallenge experiment. CR mice were rechallenged with 1000 freshly dissociated tumor cells in the contralateral mammary gland. **F** Tumor volume of weight matched control (naïve) mice and CR mice rechallenged with T12 tumor cells. Significance determined by Two-way ANOVA and Šidák's multiple comparisons test (*n* = 5 control mice and *n* = 4 CR mice). **G** UMAP visualization post 10 day (short-term) treatment of IgG Control, CTX, SNDX-ms6352 or CTX + SNDX-ms6352 in T12 and 2151 R primary tumors (*n* = 3 mice per treatment). **H** Gene ontology analysis of residual inflammatory macrophage phenotypes. **I** Gene ontology analysis of residual lipid-associated macrophage phenotypes. Differential gene expression was determined using the Wilcoxon rank-sum test (two-sided). Genes with a Log2 (fold change) >0.5 or <−0.5 and an adjusted *P*-value of <0.01 (Benjamini−Hochberg correction) were considered significant. GO pathway enrichment analyses were performed using a hypergeometric test, and an adjusted *P*-value <0.05 was considered significant. **J** Circle plot demonstrating intercellular cell-cell communication and disruption of the CSF signaling network between immune and tumor cells within each treatment. **K** Plasma cytokine levels from tumor bearing mice following each treatment in T12 primary tumors, all cytokine levels were normalized to their respective IgG Controls. Significance determined by ordinary one-way ANOVA and Tukey's multiple comparisons test (*n* = 3 mice per treatment). Data shown as mean ± SEM.

biomarker (Fig. 1K). CTX alone increased the levels of IL-17, a pro-inflammatory cytokine that has dichotomous anti- and pro-tumor effects[35,36]. Interestingly, we observed elevated levels of G-CSF following SNDX-ms6352, confirming the ying and yang between macrophages and neutrophils observed previously[6,15,37,38]. Within the tumor lysate CTX upregulated MIG but also increased RANTES (CCL5) a known macrophage recruiting cytokine[39], and this could potentially explain why tumors recur post CTX treatment cessation (Fig. S2F and Supplementary Dataset S13). Furthermore, tumor lysate cytokines from the combination treatment yielded elevated levels of IL-5 and G-CSF (Fig. S2F). Intriguingly, tumor regression was not observed when T12 tumors were grown in T cell deficient nude mice and subjected to the same treatments (Fig. S2G). Thus, we speculate that in primary tumors, the combination treatment leads to long-term anti-tumor immunity via activation of pro-inflammatory pathways including type II interferon coupled with T, B and NK cell recruitment (Fig. S2H).

## An elevated PD-1/PD-L1 axis is associated with ITH associated recurrence in lung metastases

TNBC is an aggressive highly invasive subtype with high degree of visceral metastasis including metastasis to the lung[40] and liver[41]. Both are associated with the worst prognosis and survival rates for breast cancer. Since macrophage depletion coupled with CTX disrupted the TIME which led to a sustained immunological response for primary tumors, we next determined if the combination similarly altered the TIME in lung metastatic sites. To test the long-term efficacy of the combination we designed a similar treatment schedule as the primary tumors which included treatment cessation at day 28 (Fig. 2A). Because luciferase and green fluorescent protein (GFP) are considered fluorophore-immunogenic neoantigens[42–45] we did not use them to track metastases. Thus, lung metastases were generated using 30,000 freshly dissociated (unlabeled) T12 tumor cells. Bromodeoxyuridine (BrdU) was utilized to identify (ID) established metastases post 12 day TV injection (Figs. 2B and S3A). Although, a single dose of SNDX-ms6352 efficiently depleted F4/80⁺ TAMs in lung metastasis (Fig. S4B and S4C) only single agent chemotherapy or the combination increased survival as compared to the IgG control and SNDX-ms6352. However, compared to chemotherapy alone the combination significantly prolonged overall survival (Fig. 2C). Mice that exhibit lung metastases deteriorate rapidly, and single agent treated mice lost considerable body weight compared to CTX, or combination treated mice (Fig. 2D). To study the landscape of the lung metastatic TIME we compared IgG Control lungs (Controls 1&2), and combination treated lungs (Combos 1&2) using IMC (Fig. 2E). Controls 1&2 expressed immunologically cold, highly proliferative macro-metastases (Ki67⁺, Vim⁺, CD44⁺) with infiltration of F4/80⁺ TAMs and S100A8/9⁺ neutrophils (Fig. S4D). Although Combos 1&2 expressed decreased proliferative tumor cells (Ki67⁺, Vim⁺, CD44⁺) these immunologically cold micro-metastases displayed elevated levels of exhaustion markers

PD-L1 and PD-1 (Figs. 2F−H and S4E). Intriguingly, IMC uncovered multiple micro-metastases adjacent to each other, and each had its own unique phenotype, even though spatially they were in close proximity (Figs. S4F and S4G). Thus, we next asked if the ITH observed within the metastatic TIME might account for the suppression of long-term immunity leading to recurrence. Our previous observations of B- and T cell accumulation within the mammary gland of primary tumor cell-rechallenged mice, led us to speculate that a similar pattern would occur in the metastatic setting. However, following tumor cell rechallenge, prior to any tumor growth in the mammary gland all the mice exhibited lung metastatic recurrence (Fig. 2I). Moreover, H&E and IHC identified an immunologically cold TIME with high levels of F4/80⁺ TAMs (Fig. S4H). Activation of the immune checkpoint (PD-1/PDL-1) axis can lead to tumor cell chemoresistance and is associated with increased metastasis[46].

## Addition of ICB reshapes the metastatic TIME and leads to a T cell proximity response

We, therefore hypothesized that ITH within the lung metastatic TIME which displayed PD-L1 led to a dual effect resulting in F4/80⁺ TAM accumulation and T cell repression and exhaustion. The high levels of PD-L1 may lead to T-cell exhaustion if T cells begin to infiltrate. This might account for the presence of an immunologically cold environment and eventual recurrence following tumor cell-rechallenge. PD-L1 is a well-known T cell exhaustion marker that is not commonly observed in non- metastatic breast cancer. In 30−60% of patients with metastatic TNBC, elevated levels of PD-L1 also may be a marker of an IFN-high TIME and can be associated with a favorable prognosis[47–49]. However, therapeutic intervention of this axis can re-activate exhausted T cells within the TIME and lead to anti- tumor immunity[50]. To directly compare the metastatic TIME between differing combinations which included the triple combination of CTX with SNDX-ms6352 and anti-PD-1, all lungs were harvested at day 7 post treatment (Fig. 3A). For short-term studies treatment administration was delayed until day 19 to ensure that all treated samples had a sufficient metastatic burden for analysis (Fig. 3A). Single agent and double combination treated mice lost a significant body weight as compared to mice that received additional ICB (Fig. 3B). Using a previously published size-based scoring system[51], chemotherapy and both the double and triple combination treated mice had a significantly lower metastatic score as compared to single agent treated mice (Figs. 3C, D and S5A, Supplementary Table 1). IMC analysis revealed changes to the metastatic TIME. Both double and triple combinations decreased proliferative tumor cells (Ki67⁺, Vimentin⁺ and CD44⁺) and TAMs (F4/80⁺) as compared to single agent treatments (Fig. 3E and S5B-S5D). However, other than neutrophils immune cell changes were not observed within the tumor following combination treatments, likely due to the experimental design (short term treatment) (Figs. 3E, F and S5E). Independent single cell unsupervised cluster analysis identified several

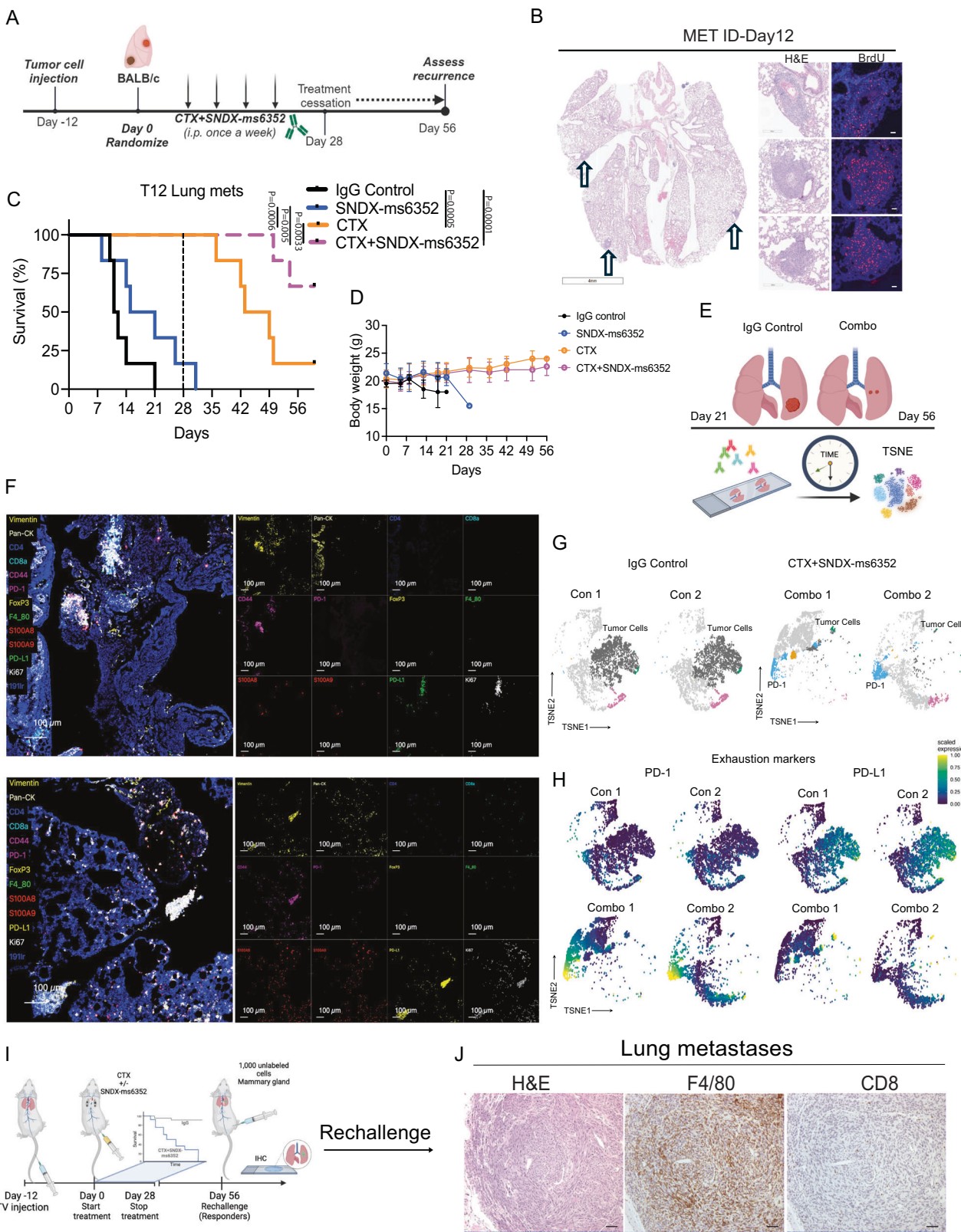

individual clusters in IgG controls that included Ki67/ vimentin while the triple combination consistently expressed CD11b/ Ly6C/ B220 and Ly6c/ CD4/ CD8 (Fig. S5E and S5F, S5G). Importantly, only the double combination treatment showed an increase in PD-L1/PD-1 as compared to the triple combination, consistent with our previous observations (Fig. 3G). Finally, the addition of ICB depleted the PD-L1/PD-1 axis in inhibitory TAMs and increased the number and proximity of T cells

and neutrophils to the tumor bed driving an anti-tumor response (Fig. 3H). PD-L1 is often upregulated in response to stress or inflammation. However, we speculate that the levels of PD-L1 may decrease as the metastatic burden diminishes in response to a shift within the TIME leading to fewer PD-L1+ Vim+CD44+ tumor cells[52,53]. These outcomes are consistent with the dynamic regulation of PD-1/PD-L1 in response to changes in tumor burden and immune tone.

**Fig. 2 | Combination treatment leads to ITH driven PD-1/PD-L1 axis in metastatic lung metastases. A** Schematic illustration of treatment regimen. Tumor cells are first injected via TV into female Balb/c mice. Post 12 day TV injection the mice were randomized and considered to be day 0. IgG control, CTX and SNDX-ms6352 were administered i.p. once a week for 4 weeks. All treatments were stopped at day 28 and recurrence was assessed at day 56. **B** Representative H&E and immunofluorescence staining of BrdU from an independent lung bearing metastases 12 days post TV injections (Representative image of *n* = 2 mice). **C** Kaplan-Meier survival curves following IgG Control, SNDX-ms6352, CTX and CTX + SNDX-ms6352 in T12 with established lung metastases. Statistical analysis was performed using the Log-rank (Mantel-cox) test (*n* = 6 mice per treatment). **D** Mouse body weight (g) following IgG Control, SNDX-ms6352, CTX and CTX + SNDX-ms6352 in T12 established lung metastases. Significance determined by Two-way ANOVA and Tukey's multiple comparisons test (*n* = 6 mice per treatment). **E** Schematic illustration of IgG control mice at day 21 and CTX + SNDX-ms6352 treated mice at day 56, subjected for IMC analysis to study the metastatic TIME. **F** IMC analysis post CTX + SNDX-ms6352 treated T12 model at day 56 from independent lung tissues. Representative markers included Vimentin, Pan-CK, CD44, S100A8/9, PD-L1, PD-1, Ki67, and 191ir (DAPI). **G** TSNE plot visualization of tumor cells, PD-L1, PD-1 and Pan-CK in IgG control and CTX + SNDX-ms6352 treated. **H** TSNE plot visualization of PD-L1 and PD-1 expression in IgG control and CTX + SNDX-ms6352 treated. **I** Schematic illustration of tumor cell re-challenge. Mice that were deemed as complete responders (CR) at day 56 were re-challenged by injecting 1000 unlabeled cells into the mammary gland (primary site of tumor origin). **J** H&E and IHC staining of macrophages (F4/80) and T cells (CD8) in recurrent lung metastases (Representative image of *n* = 4 mice). Data shown as mean ± SEM.

## Triple combination leads to a long-term anti-tumor response of established lung metastases

To assess the long-term efficacy of SNDX-ms6352 and CTX with ICB on established lung metastases, we repeated these initial experiments and injected 30,000 freshly dissociated (unlabeled) T12 tumor cells into the TV. To mimic what recently has been introduced in the clinic, we included a separate group that received a maintenance dose of ICB post 28 day treatment cessation[54]. In this experiment the mice were monitored for up to 70 days to assess recurrence (Fig. 4A). Single agent therapies, IgG control and anti-PD-1 had modest effects compared to the double and triple combination (Fig. 4B). However, only the mice that continued to receive a maintenance dose of anti-PD-1 post treatment cessation had better survival (Fig. 4B). The mice that received the triple combination tolerated the maintenance dose and preserved body weight, even when treated for up to 70 days (Fig. 4C). To test systemic innate immunity, the mice were rechallenged by injecting tumor cells into the mammary gland. Intriguingly the rechallenged mice impeded primary tumor growth as compared to naïve control mice (Fig. 4D). IF staining confirmed a decrease of BrdU+ tumor cells, loss of CSF-1R signal and infiltration of CD20+ B and CD8+ T cells following the double combination treatment in lung metastases (Fig. 4F and S6A). Although we did detect increased levels of CD20 and CD8 in the double and triple combination, residual tertiary lymphoid structures (TLS) were only observed in the triple combination within the lung. (Fig. 4E, F). Further, IHC confirmed loss of F4/80+ TAMs and accumulation of S100A8+ neutrophils and CD4+, CD8+ T cells within lung metastases following the triple combination (Fig. S6B and S6C).

## Immunologically cold T12 liver metastases are highly infiltrated by TAMs

Patients diagnosed with TNBC liver metastases are associated with significant morbidity and have the worst overall survival compared to other breast cancer subtypes with liver metastases[55,56]. These patients also exhibit lower immunotherapy efficacy via macrophage T cell killing mechanisms[57]. We speculated that the combination designed to eliminate TAMs with SNDX-ms6352 coupled to CTX and ICB that potentiates T cells in lung metastases might also be efficacious in the treatment of liver metastases. Thus, we proceeded to generate organ specific liver metastases by injecting 5000 freshly dissociated (unlabeled) T12 tumor cells into the portal vein (PV)[58] (Fig. 5A). Intriguingly, the liver metastases were highly infiltrated by F4/80+ TAMs and S100A8+ neutrophils and devoid of CD8+ T cells, while CD4+ T cells remained spatially distant from the metastases (Fig. 5B). H&E and BrdU staining were utilized to identify established metastases post 15-day PV injection (Fig. 5C). To study the metastatic TIME of liver metastases all liver tissues were harvested at day 5-7 post IgG Control, SNDX-ms6352, CTX and CTX + SNDX-ms6352 +/-anti-PD-1 (Fig. 5D). Only the triple combination treated mice, maintained body weight (Fig. 5E) and both the double and triple combination treated mice exhibited a lower metastatic score compared to single agent treated mice (Figs. 5F,

5G and S7A, Supplementary Table 2). Compared to IgG control, SNDX-ms6352 decreased F4/80+ TAMs and S100A8/9+ neutrophils. CTX induced CD4/CD8+ T cells and upregulated Pan-CK, F4/80+ TAMs and PD-L1 (Figs. 5H, 5I and S7B). In contrast, the double combination increased PD-1 the triple combination displayed modest elevated levels of CD4, Pan-CK and PD-L1 (Figs. 5H–J and S7B). However, unlike single agents including CTX both double and triple combinations decreased Ki67+, Vimentin+ proliferative tumor cells and F4/80+ TAMs. Single cell unsupervised cluster analysis identified several individual clusters similar to the lung metastatic TIME. IgG controls displayed elevated Ki67/Vimentin/CD44 and CD44/F4/80 while the triple combination had clusters that consistently displayed S100A8/9 with B220 B cells and CD4/CD8a T cells (Fig. S7C and S7D). Loss of F4/80 TAMs was accompanied by the close proximity of CD4/CD8+ T cells to tumor cells following the triple combination (Fig. S7E). In addition, Ly6G+ S100A8/9+ neutrophils surrounded the tumor cells (Fig. S7E and S7F). Furthermore, SNDX-ms6352 efficiently depleted F4/80+ TAMs within the liver metastatic tumor bed without affecting F4/80+ Kupffer cells in the surrounding liver stroma (Fig. S7G).

## Addition of ICB maintenance dose is necessary to generate adaptative anti-tumor immunity for established liver metastases

To assess the long-term efficacy of the triple combination along with a maintenance dose of anti-PD-1 on established liver metastases, we repeated these experiments and injected 5000 freshly dissociated (unlabeled) T12 tumor cells into the PV. Similar to our previous experiment we assessed recurrence for up to 56 days post 28 day treatment cessation (Fig. 6A). For liver metastases we used PET/CT to detect fluorine-18 labeled fluoroxyglucose ($^{18}$F-FDG) which identified and confirmed established liver metastases post 15 day PV injections (day 0) (Fig. 6B). The same mice were imaged at day 56 (post-treatment) and had visibly and quantitatively less liver metastases following CTX + SNDX-ms6352+aPD-1 (m.dose) (Fig. 6C). CTX and double combination-treated mice, significantly increased overall survival compared to IgG control and SNDX-ms6352-treated mice, however, only mice that received the triple combination had better survival (Fig. 6D). Remarkably 66% of the mice that received an ICB maintenance dose post treatment cessation achieved a CR (Fig. 6D). Although statistically not significant, 75% as compared to only 33% of the mice died when the maintenance dose was not given. Further, CTX, double and triple combination treated mice preserved body weight for up to 56 days (Fig. 6E). IF staining demonstrated decreased levels of CSF-1R following SNDX-ms6352 compared to IgG control. However, in liver metastases only the triple combination displayed increased CD20+ B and CD8+ T cells (Figs. 6F, G and S8A). IHC confirmed diminished levels of F4/80+ TAMs following SNDX-ms6352, and in the double and triple combination-treated mice. Importantly only the triple combination displayed high CD8+ and CD4+ T cells and S100A8+ neutrophils consistent with our previous observations

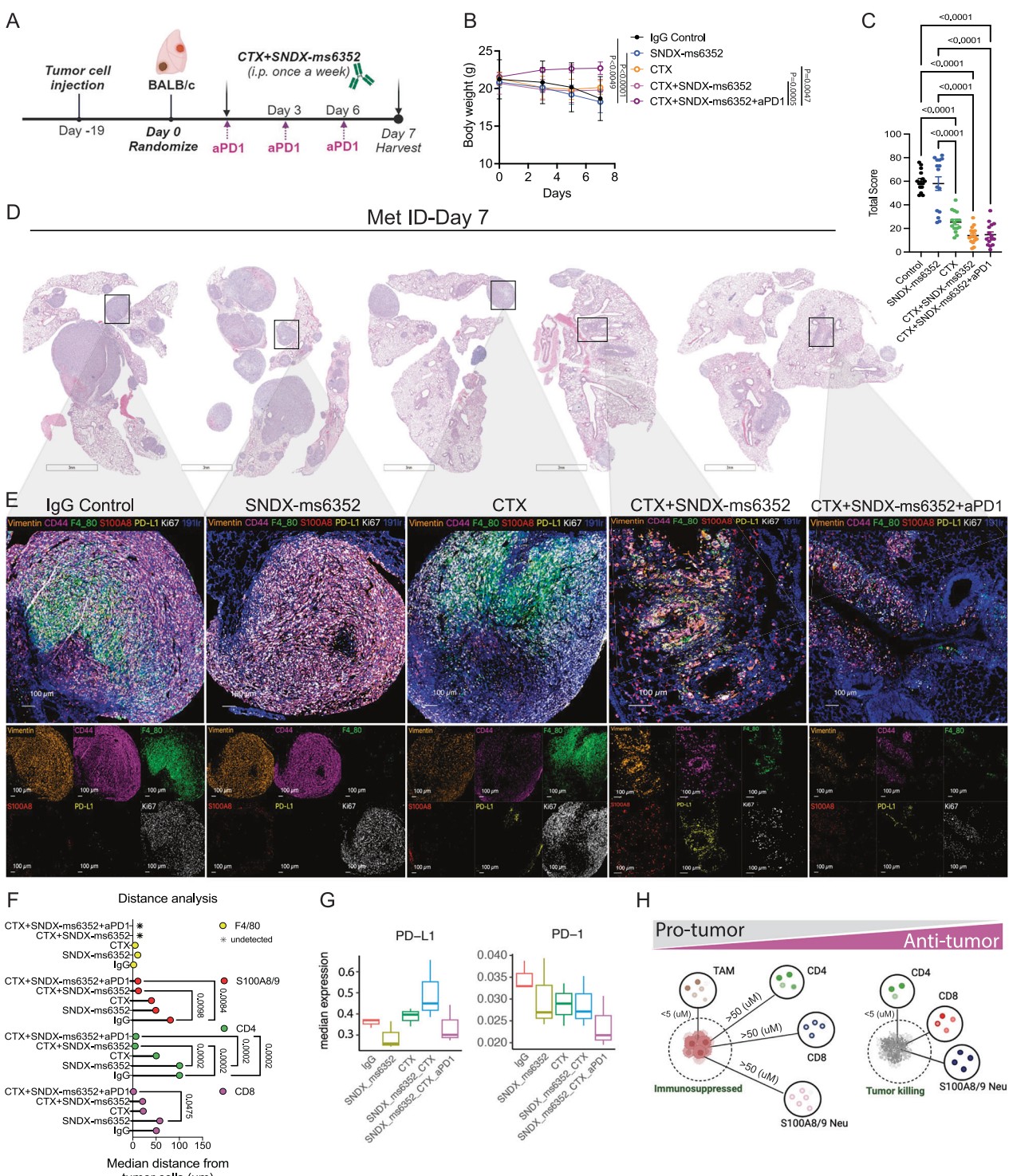

(Fig. S8B and S8C). Impressively all the mice that received the maintenance dose cleared the mammary gland tumor cell rechallenge (Fig. 6H and S8D). To further test the adaptive immunity of these mice we isolated splenocytes (CD3+ T cells) and performed an adoptive T cell transfer (ACT) into T12 tumor bearing mice (Fig. 6I). We did not observe any liver metastases following the liver rechallenge and thus we deemed these mice "cured" as depicted in our schematic (Fig. 6I), prior to performing the adoptive T cell transfer from the same mice. Because of the possibility of immediate T cell exhaustion, we tested if a single dose of anti-PD-1 would benefit the ACT. Intriguingly the addition of anti-PD-1 with T cells significantly slowed the growth of tumor bearing mice compared anti-PD-1 and T cells alone (Fig. 6J and S8E).

These mice also had significantly smaller spleens compared to mice that received IgG control, anti-PD-1 and T cells alone (Fig. 6K) and this correlated with a positive outcome[59,60]. The spleens and tumors of the mice that received anti-PD-1 treatment along with T cells contained higher levels of CD8+ T cells (Figs. 6L, M and S8F, S8G). The rechallenged mice displayed systemic anti-tumor immunity and ACT donor mice most likely benefited from central memory T cells.

## TNBC patients display elevated immunosuppressive macrophage signatures in metastatic sites

In the metastatic setting our data suggest that our models resemble an immunologically cold TIME with an abundance of TAMs that are

**Fig. 3 | Addition of anti-PD1 reshapes the landscape of the metastatic TIME leading to the proximity of B-, T cells and neutrophils and a therapeutic response in established lung metastases. A** Schematic illustration of short-term treatment regimen. **B** Mouse body weight (g) following short-term treatment of IgG Control, SNDX-ms6352, CTX, CTX + SNDX-ms6352 and CTX + SNDX-ms6352 + aPD1 in T12 models with established lung metastases. Statistical analysis was performed using Two-way ANOVA and Tukey's multiple comparisons test (*n* = 5 mice per treatment). **C** Quantification of lung metastases post short-term treatment. Serial sectioning was performed to collect a total of 10 sections for each sample. Metastases with sizes of <1 mm, 1–3 mm, 3-5 mm, and >5 mm were assigned scores of 1, 2, 3, and 4, respectively. Significance determined by Two-way ANOVA and Tukey's multiple comparisons test (*n* = 15 independent ROI). **D** Representative H&E staining of lung metastases at day 7 post IgG control, SNDX-ms6352, CTX, CTX + SNDX-ms6352 and CTX + SNDX-ms6352 + aPD1 treatments (*n* = 5 mice per treatment). **E** IMC analysis post IgG control, SNDX-ms6352, CTX, CTX + SNDX-ms6352

and CTX + SNDX-ms6352 + aPD1 treated T12 model at day 7. **F** Median distance of immune cells from tumor cells post IgG control, SNDX-ms6352, CTX, CTX + SNDX-ms6352 and CTX + SNDX-ms6352 + aPD1, distance analysis is representative from IMC analysis of individual channels CD8, CD4, S100A8/9 and F4/80. Significance was determined using Two-way ANOVA and Tukey's multiple comparisons tests (*n* = 3 ROI). **G** Median expression levels of the exhaustion markers post IgG control, SNDX-ms6352, CTX, CTX + SNDX-ms6352 and CTX + SNDX-ms6352 + aPD1 treated T12 model at day 7. Box plots display the median (center line), interquartile (25th -75th percentiles) and whiskers extending to the minimum and maximum values, outliers are shown as individual points. *P*-values were generated using diffcyt function (type of analysis, DS; method for DS testing, diffcyt-DS-limma). *P*-values were then adjusted using the Benjamini-Hochberg method (*n* = 3 ROI). **H** Illustration depicting pro-tumor immunosuppressed and anti-tumor, tumor killing phenotypes. Phenotypes are representative of the proximity between TAMs, CD4, CD8, S100A/9 Neutrophils within tumor cells. Data shown as mean ± SEM.

specifically devoid of T cells. Thus, we analyzed the transcriptomic dataset of TNBC patients from the AURORA clinical study (**GSE209998**) which genomically profiled matched metastatic and primary breast cancers (Fig. 7A)[11]. We determined if any clinical correlations existed between the metastatic TIME of patients and that of T12 lung and liver metastases using multiple immune subsets and specific macrophage signatures. CIBERSORTx signatures revealed an overactive TIME that upregulated multiple T cell (CD3, CD8 and CD4), natural killer (NK) cell, memory B and CD19 B cell, as well as several monocyte and macrophage immunity (M0 and M1) signatures compared to matched lung or liver metastases (Fig. 7B). This correlated in patients with varying breast cancer subtype (LumA, LumB and HER2E) (Fig. S9A). However, lung and liver metastatic sites exhibited increased claudin-low associated gene signatures with several patients having signatures that phenocopy the p53 null mouse models (MM) (Fig. 7B). Individual gene expression profiling identified significantly increased immunosuppressive macrophage signatures, *CSF1R*[61], *MRC1*[62] and *TREM2*[63] in metastases compared to primary tumors (Fig. 7C). Unbiased gene expression profiling between metastatic sites vs primary tumors showed upregulation of the known TAM marker *CD163*[64,65] and *CXCL11* a macrophage-derived chemokine[66–68]. More in-depth analysis between organ specific metastases revealed upregulation of the transcription factor hes family BHLH transcription factor 2 (*HES2*) in lung vs breast, the p53 degrader ubiquitin D (*UBD*) in liver vs breast and the preferentially expressed antigen of melanoma (*PRAME*) gene in liver vs lung, which has been observed in metastatic breast cancers (Fig. S9B)[69]. Although, gene expression profiles will never be identical between murine models and humans, in this study we showed that similar immune profiles, specifically macrophage associated genes do exist between the metastatic TIME of murine models and that of human metastatic samples. In summary, our study demonstrates that administration of ICB along with combination anti-CSF1R and low-dose CTX therapy was able to reshape the metastatic landscape of lung and liver metastases resulting in increased TILs with B cells and neutrophils, prolonged survival and a CR (Fig. 8).

## Discussion

TAMs have been readily observed in multiple types of cancers including breast and are associated with poorer response to both chemo- and immunotherapies[8,70–74]. In breast cancers a correlation between TAMs and TILs exists, with the former being highly upregulated compared to the latter[8,75]. Preclinical efforts to target TAMs in TNBC have previously demonstrated positive outcomes. For instance, anti-CSF1R antibody therapy successfully reduced F4/80⁺ TAMs to overcome PARPi resistance in a BRCA1-deficient TNBC model mediated by CD8⁺ T cells[76]. These results coincide with our previous findings where we showed a B- and T cell response using a small molecule inhibitor of CSF-1R, PLX-3397 in primary p53 null syngeneic models[15].

While PLX-3397 inhibits CSF-1R, it also exhibits activity against c-Kit and FLT3[61]. These off-target effects may in part contribute to the liver toxicity that resulted in PLX-3397 being discontinued in the iSPY2 trial for the treatment of TNBC. However, the absence of liver toxicity of Axatilimab-csfr has, been demonstrated in patients, and Niktimvo is now FDA approved for graft versus host disease and is being studied in several breast cancer clinical trials (NCT06488378 and NCT05491226).

Therefore, to efficiently target TAMs and limit toxicities we tested the CSF-1R antibody SNDX-ms6352 (Axatilimab-csfr)[22]. While SNDX-ms6352 depleted F4/80⁺ TAMs it did not halt tumor growth, similar to what was observed with PLX-3397. In this study we combined SNDX-ms6352 with a low-dose of CTX which has been shown to be immunomodulatory in both murine models and patients[77–79]. Preliminary studies from our laboratory revealed that that CTX reshapes the myeloid compartment of the bone marrow, ablating neutrophils and increasing monocytes locally and systemically. Further, low-dose CTX was selected specifically for its immunomodulatory effects—particularly its documented ability to selectively deplete regulatory T cells (Tregs), thereby potentially enhancing anti-tumor immunity in combination with immunotherapy agents. As previously reported low-dose CTX has demonstrated efficacy in modulating the immune microenvironment, promoting T-cell-mediated responses, and reducing immunosuppressive barriers within the tumor milieu[80,81]. Interestingly, following CTX treatment cessation we observed increased TAMs within the tumor bed. Therefore, only the combination treatment yielded a CR in the T12, 2151 R claudin-low and 2336 R basal-like models. Although 2151 R was sensitive to CTX, tumor cell-rechallenged mice failed to reject tumor growth and lacked B- and T cell infiltration due to the absence of the adaptive immunity required to eliminate the cell rechallenge that was only provided by the combination treatment. Moreover, we have previously reported that when we deplete the macrophages there is a significant increase in neutrophil recruitment[6] compensating for the loss of those macrophages thus making the tumors like 2208 L less responsive. In the responders single-cell transcriptomics and cytokine profiling confirmed T- and B cell infiltration with loss of immunosuppressive *Csf1r* and *Trem2* TAMs via increased levels of IL-5, IFN-y, IL-4 and MIG. Interestingly residual inflammatory/ lipid associated T12 TAMs expressing *Fabp5* involved in lymphocyte activation and cytokine mediated pathways were observed. Fabp5 is known to antagonize immunosuppressive macrophage related markers *Fizz1, CD206* and *Arg1* and is important for the maintenance and survival of CD8⁺ T cells[82–84]. Successful CSF-1R inhibition[85] was indicated by the elevated expression of the M-CSF biomarker in the plasma. Mechanistically the residual inflamed-TAM (MHC II⁺) and previously observed Fabp5⁺ LA-TAMs coordinate a long-term anti-tumor response by attracting CD4⁺/CD8⁺ T and B cells. Within the combination treated tumors, CD4 T and dendritic cell increased interferon gamma receptor 1, the specific receptor for IFN-y, while

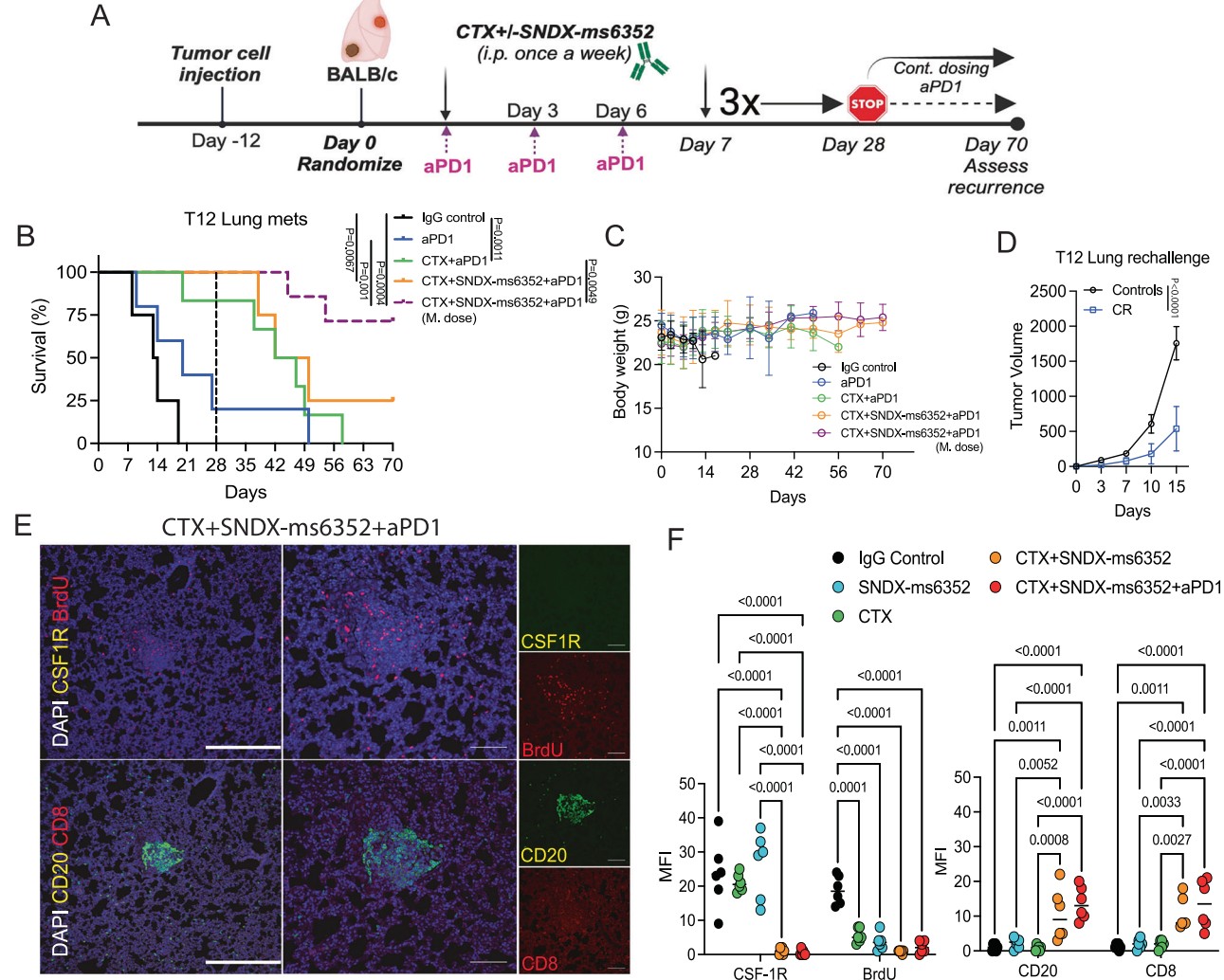

**Fig. 4 | Triple combination treatment leads to long-term durable response of established lung metastases. A** Schematic illustration of long-term treatment regimen. T12 tumor cells are first injected via TV into female Balb/c mice. Post 12 day TV injection the mice were randomized and considered to be day 0. IgG control, CTX, SNDX-ms6352, CTX + SNDX-ms6352 and CTX + SNDX-ms6352 + aPD1 were administered i.p. once a week for 4 weeks, anti-PD-1 and its IgG2a isotype control were administered every 3 days. Treatments were stopped at day 28 and recurrence was assessed at day 56, except for the group that continued to receive a maintenance does (M. dose) of anti-PD-1. **B** Kaplan-Meier survival curves following IgG Control, SNDX-ms6352, CTX, CTX + SNDX-ms6352 and CTX + SNDX-ms6352 + aPD1 +/- M. dose in T12 syngeneic model with established lung metastases. Statistical analysis was performed using the Log-rank (Mantel-cox) test ($n$ = 5-7 mice per treatment). **C** Mouse body weight (g) following IgG Control, SNDX-ms6352, CTX, CTX + SNDX-ms6352 and CTX + SNDX-ms6352 + aPD1 +/- (M. dose) in

T12 syngeneic model with established lung metastases. Significance was determined by Two-way ANOVA and Tukey's multiple comparisons test ($n$ = 5 mice per treatment). **D** Tumor volume of weight matched control (naïve) mice and CR mice rechallenged with T12 tumor cells. Significance was determined by two-way ANOVA and Šidák's multiple comparisons test ($n$ = 5 control mice and $n$ = 5 CR mice). **E** Double immunofluorescence (IF) staining of CSF-1R, BrdU, CD20, CD8 and DAPI in T12 syngeneic model with established lung metastases treated with CTX + SNDX-ms6352 + aPD1 (Representative image of $n$ = 5 mice). **F** Quantification of mean fluorescence intensity (MFI) staining of CSF-1R, BrdU, CD20 and CD8 in T12 mice with established lung metastases following IgG control, CTX, SNDX-ms6352, CTX + SNDX-ms6352 and CTX + SNDX-ms6352 + aPD1. Significance was determined by Two-way ANOVA and Tukey's multiple comparisons test ($n$ = 6 independent ROI). Data shown as mean ± SEM.

SNDX-ms6352 alone downregulated CSF-1R in both monocytes and macrophages (Fig. S3E). These results were confirmed by IHC whereby SNDX-ms6352 efficiently depleted F4/80, CSF-1R and the immunosuppressive macrophage marker ARG1 in primary tumors (Fig. S3F).

Although, the treatment outcomes for primary tumors yielded impressive results, most TNBC patients die from metastatic disease[9] with metastatic TNBC having a 5 year survival rate of 11% and a median survival of 11–13 months[86]. Thus, we tested the efficacy of the double combination treatment on two of the most prevalent sites of metastasis for TNBC, the lung and liver[40,41]. Women who present to the clinic have established metastatic lesions; thus, we waited about 2 weeks post TV or PV injections to mimic what is being observed in the clinic. Prior to treatment H&E and BrdU incorporation were employed to

determine the metastatic burden and ensure the presence of established lung or liver metastases. The metastatic TIME of the lungs displayed high levels of Vimentin[+], CD44[+], tumor cells and F4/80[+] TAMs. Although, the double combination was well tolerated, and several mice reached CR after 56 days. ITH was observed within the metastatic TIME displaying de-novo levels of PD-L1 within double combination treated mice followed by rapid recurrence of lung metastases. Furthermore, following tumor cell rechallenge these mice failed to generate systemic adaptive anti-tumor immunity. Intra-patient ITH (multiple nodules within the same individual patient) are associated with lower response rates, treatment insensitivity and resistance[87,88].

Addition of ICB to SNDX-ms6352 and CTX resulting in sustained loss of TAMs, tumor cells and decreased PD-L1/PD-1 expression

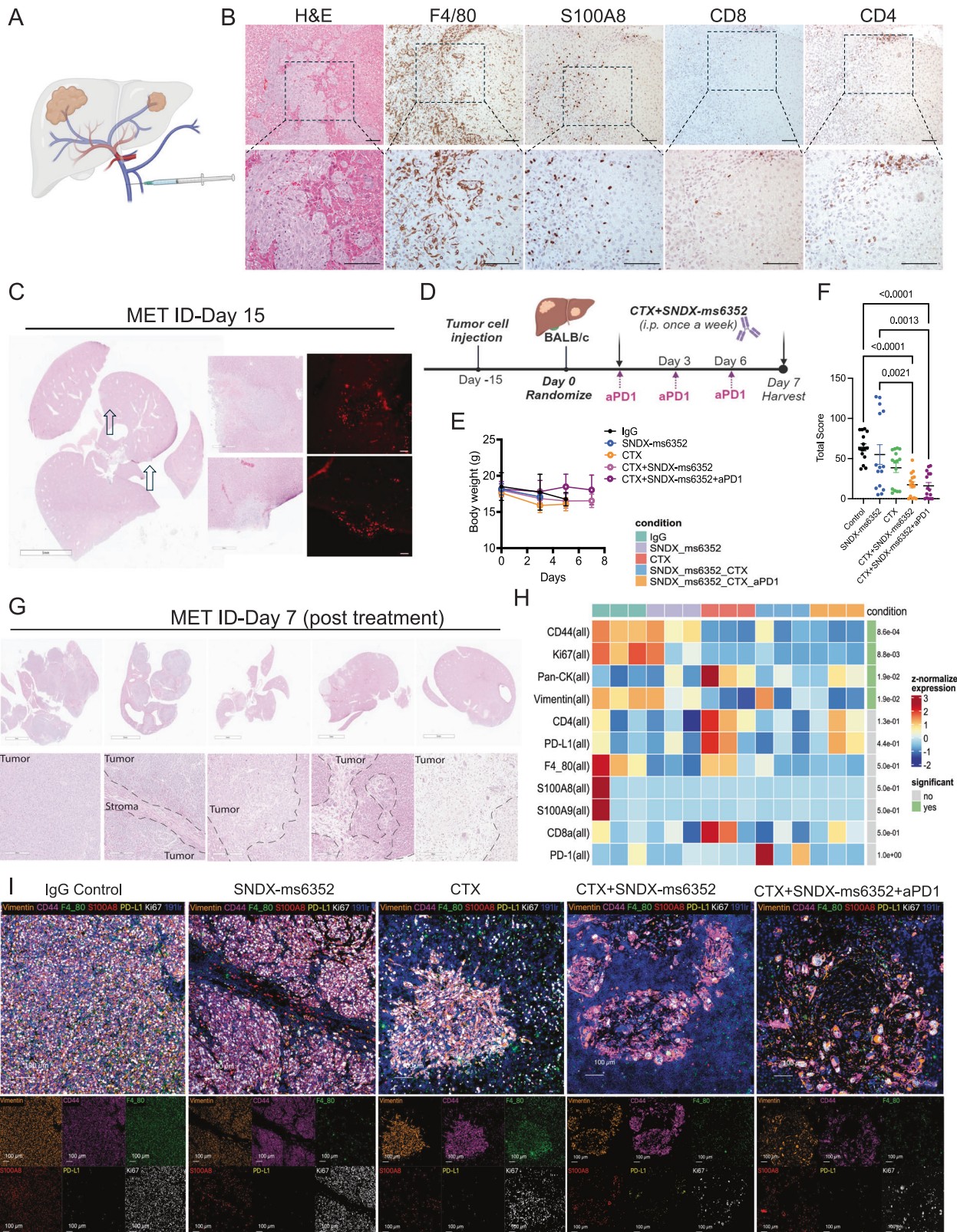

accompanied by B- and T cell residual TLSs. Established liver metastases exhibited an immunologically "cold" TIME. Although, we did not observe TLSs in the liver, the triple combination did increase B- and T cells. Impressively, this triple combination eradicated established liver metastases. Furthermore, systemic adaptive anti-tumor immunity was observed when T cells reduced tumor burden following ACT. Importantly SNDX-ms6352 targeted F4/80$^+$ TAMs within the

metastatic tumor bed but spared the majority of Kupffer cells within the liver. In clinical trials, the observed effects of Axatilimab-csfr which increased liver enzymes may be attributed to a decrease in Kupffer cells; however, these changes did not lead to organ damage[22]. To evaluate the potential clinical significance of these findings, we analyzed results from the AURORA trial, one of the most up to date and unique breast cancer specific clinical studies that encompasses

**Fig. 5 | Established T12 liver metastases present with an immunologically cold TIME. A** Schematic illustration of PV injection utilized to generate experimental liver metastases. **B** Representative H&E and IHC staining of F4/80, S100A8, CD8 and CD4 in liver metastases post 15 days PV injections (Representative image of $n = 3$ mice). **C** Representative H&E and IF staining of BrdU in liver metastases 15 days post T12 tumor cell PV injections (Representative image of $n = 2$ mice). **D** Schematic illustration of short-term treatment regimen. **E** Mouse body weight (g) following short-term treatment of IgG control, SNDX-ms6352, CTX, CTX + SNDX-ms6352 and CTX + SNDX-ms6352 + aPD1 in T12 syngeneic model with established liver metastases. Statistical analysis was performed using Two-way ANOVA and Tukey's multiple comparisons test ($n = 3$ mice per treatment). **F** Quantification of liver metastases post short-term treatment. Serial sectioning was performed to collect a total of 10 sections for each sample. Metastases with sizes of <1 mm, 1–3 mm, 3–5 mm, and >5 mm were assigned scores of 1, 2, 3, and 4, respectively.

Statistical analysis was performed using Two-way ANOVA and Tukey's multiple comparisons test as described above ($n = 15$ independent ROI). **G** Representative H&E staining of liver metastases at day 7 post IgG control, SNDX-ms6352, CTX, CTX + SNDX-ms6352 and CTX + SNDX-ms6352 + aPD1 treatments ($n = 3$-5 mice per treatment). **H** Heatmap depicting median expression levels of various immune, tumor, T-cell exhaustion and proliferative markers following IgG control, SNDX-ms6352, CTX, CTX + SNDX-ms6352 and CTX + SNDX-ms6352+aPD-1 treated T12 liver metastases from IMC analysis. *P*-values were generated using diffcyt function (type of analysis, DS; method for DS testing, diffcyt-DS-limma). *P*-values were then adjusted using the Benjamini-Hochberg method ($n = 3$ ROI). **I** Cumulative and individual marker IMC analysis from IgG control, SNDX-ms6352, CTX, CTX + SNDX-ms6352 and CTX + SNDX-ms6352 + aPD1 treated liver metastases. Data shown as mean ± SEM.

primary tumors with matched metastatic samples. Transcriptomic datasets of TNBC patients from the AURORA metastatic clinical study (**GSE209998**) which genomically profiled matched metastatic and primary breast cancers[11], show upregulated T cell natural killer cell, memory and B cell, monocyte and macrophage CIBERSORTx signatures in primary tumors as compared to matched lung or liver metastases. Importantly, patients with metastatic disease had increased immunosuppressive macrophage signatures compared to primary tumors and displayed a colder TIME specially in the liver. Within the limitations of these cross-species analyses, we observe similar immune profiles, specifically an immunologically cold TIME does exist between lung and liver metastases of murine models and that of human metastatic samples. Spatially resolved metastatic breast cancer maps, similarly, identified macrophages as the most abundant immune cells, specifically in patients that had not received ICB[89,90].

Although both lung and liver metastases exhibited similarly low immune profiles, the liver was immunologically colder, which mirrors what has been observed in the AURORA clinical studies. Interestingly, in primary tumors T12 is devoid of S100A8/9+ neutrophils. However, S100A8/9+ neutrophils were now observed within the boundary of tumor metastases, and in liver metastases S100A8/9+ neutrophils were observed within the tumor bed near F4/80+ TAMs. Interestingly, we observed that Ly6G+ S100A8/9+ neutrophils post triple combination surrounded Vim+ CD44+ tumor cells within the liver. We speculate that although neutrophils can support pro-tumorigenic functions, they can also directly target tumor cells[91], therefore potentially converting them from tumor supporting to tumor-inhibitory TANs. Further studies are required to support this hypothesis.

Patients diagnosed with mTNBC will often continue to be treated for the remainder of their lives[92]. The standard of care (SOC) for mTNBC now often includes immunotherapy[93,94] with the addition of anti-PD-1[95,96]. Accordingly in the current study we demonstrated that a maintenance dose of ICB post treatment cessation of SNDX-ms6352 and CTX may be important to minimize the toxic side effects of CTX[54] and maximize treatment efficacy. Based upon these preclinical studies, this combination treatment study will now be tested in the clinic to treat patients with mTNBC, (NewSTART clinical trial (NCT06959537).

The T12 model was chosen for the experimental metastasis studies because it represents both the TNBC patients with the poorest prognosis and represents the high macrophage patient population that will be screened for the NewSTART clinical trial via pre-treated biopsies. Due to the highly aggressive nature and local invasion of the T12 claudin-low model, we have been unable to generate spontaneous metastatic models following resection. Thus, for this study we relied on experimental metastases via TV and PV injections. This approach generated organ specific metastases and allowed sufficient numbers of mice with a metastatic burden for control and treatment groups. It is not feasible to perform these treatment studies in established spontaneous

metastasis models. Previous studies from our laboratory analyzing experimental and spontaneous lung metastases using two different TNBC models showed a remarkable overlap of gene expression profiles[97]. Furthermore, we acknowledge that in patients, metastases can be multi-site and associated with higher tumor burden. This may influence the TIME, potentially altering therapeutic efficacy. In our study several mice had to be ethically euthanized due to tumor growth within the tails and these results were omitted.

Finally, although previous studies have used a variety of antibodies targeting CSF-1R (NCT01494688, NCT02760797, NCT02435680, NCT02323191) and in some cases combined these with either anti-PD-1, anti-PD-L1 or other chemotherapy agents. The combination of immunostimulatory CTX with anti-CSF-1R and anti-PD-1 is unique and was required to promote long term durable immunity in lung and liver metastases. As shown single agents targeting CSF-1R or anti-PD-1 were ineffective. This highlights the importance of appropriate combinatorial therapies and the need to test them on established metastasis.

## Methods

### Mouse models

The procedures used for in vivo animal models were conducted following protocol AN-504 approved by the Baylor College of Medicine Institutional Animal Care and Use Committee (IACUC). The ethical endpoint was met when the tumor reached a volume of 1500 mm³. For experimental metastases, body weight was measured every 3 to 7 days. The ethical endpoint was met when mice either lost ~20% body weight or display poor body condition. The mouse strains utilized, included female BALB/c mice obtained from Inotiv (strain 047) and female Hsd:Athymic Nude-*Foxn1nu*; from Inotiv (strain 069). All animals were housed in the Transgenic Mouse Facility at Baylor College of Medicine with a 12 h day/12 h night cycle in climate-controlled conditions. This study exclusively used female mice to model TNBC in women only.

### Cell lines

The *Trp53*-null TNBC syngeneic models were previously generated by transplantation of *Trp53*-deleted donor mammary epithelium into the cleared mammary fat pad of syngeneic BALB/c hosts these gave rise to re-transplantable heterogeneous mammary tumor TNBC cell lines. TNBC cell lines include T12 (BALB/c and Hsd:Athymic Nude-*Foxn1nu*), 2151 R (BALB/c), T11 (BALB/c), 2336 R (BALB/c) and 2208 L (BALB/c). Established cells were cultured in DMEM/F-12 medium (Thermo Fisher Scientific, 11330032) supplemented with 10% fetal bovine serum (FBS) (GenDEPOT, F0900-050), 5 μg/mL insulin (Sigma-Aldrich, I-5500), 1 μg/mL hydrocortisone (Sigma-Aldrich, H0888), 10 ng/mL epidermal growth factor (Sigma-Aldrich, SRP3196), and 1x Antibiotic-Antimycotic (Thermo Fisher Scientific, 15240062). All cells were tested to be free of mycoplasma contaminants using the Universal Mycoplasma Detection Kit (ATCC, 30-1012 K). Established cell lines were then transplanted

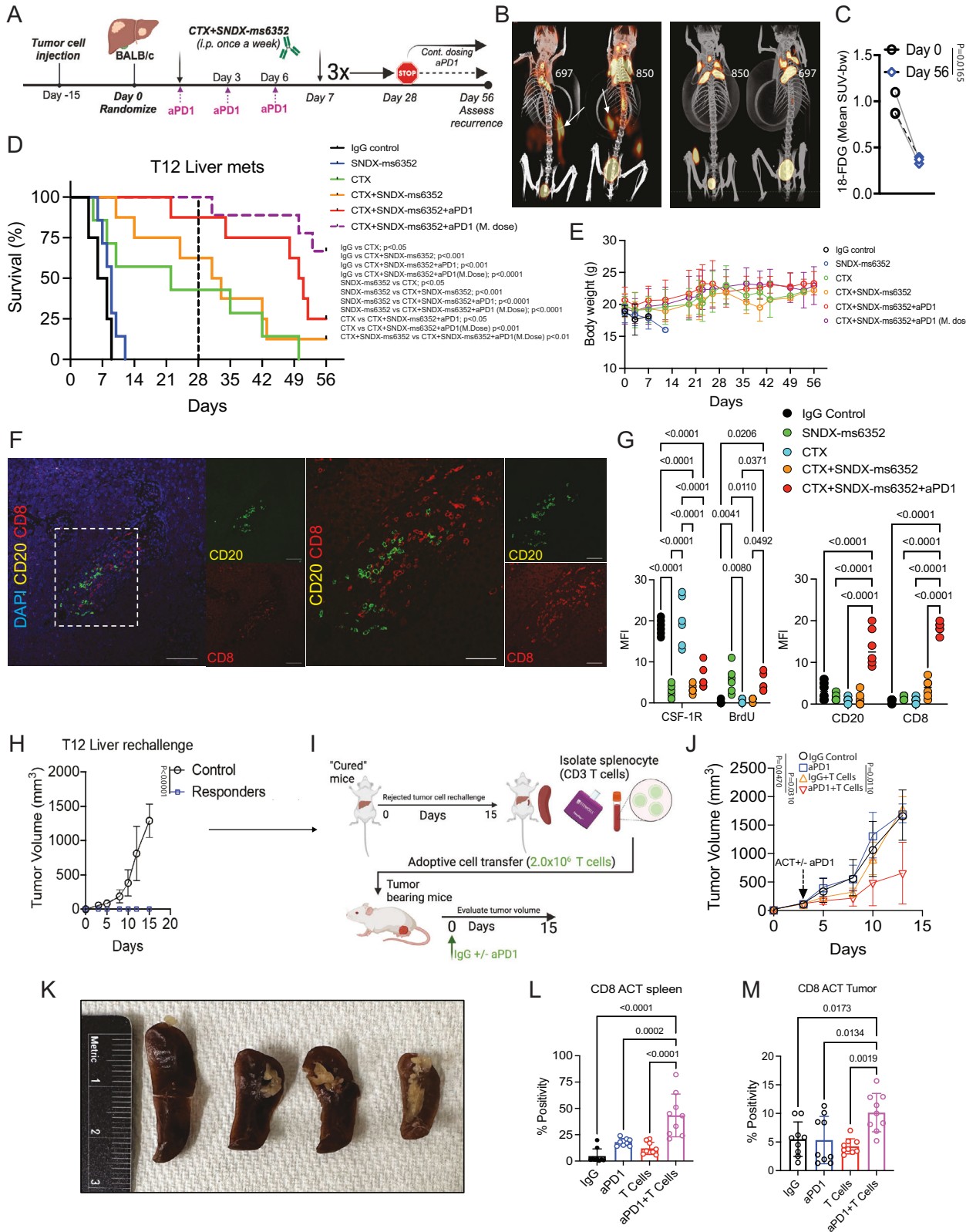

into BALB/c mice, these tumors were then banked down for long-term storage in FBS + 10% DMSO.

## Tumor transplantation

Prior to tumor transplantation and surgery, tumors were rapidly thawed and washed in PBS to remove residual DMSO. Briefly, tumor tissues were transplanted into the fourth mammary fat pad of

6–8 week-old female BALB/c from Inotiv (strain 047) mice or athymic nude mice from Inotiv (strain 069), respectively. When tumors reached an average size of 70 to 150 mm³, mice were randomized. Tumor volume and body weight were measured every 3–7 days, volume was calculated as (length × width²)/2. The ethical endpoint was met when the tumor reached a volume of 1500 mm³. Investigators were not blinded to the group assignment.

**Fig. 6 | An anti-PD-1 maintenance dose is necessary to generate long-term anti-tumor immunity and to eradicate established liver metastases. A** Schematic illustration of long-term treatment regimen. **B** Coronal view of PET/CT scans post day 56, (Representative image of $n = 3$ mice per treatment). **C** Relative quantification of 18-FDG uptake in triple combination (m.dose) treated and untreated mice. Each dot represents the mean standard uptake values (meand SUV-bw) of 18-FDG. Significance was determined by paired $t$-test ($n = 3$ mice per treatment). For PET/CT analysis, a paired $t$-test (two-sided) was used to compare groups. **D** Kaplan-Meier survival curves following various treatments including the +/- M. dose. Statistical analysis using log-rank (Mantel-cox) test ($n = 5$-$9$ mice per treatment). **E** Mouse body weight (**g**). Significance was determined by Two-way ANOVA and Tukey's multiple comparisons test ($n = 5$ mice per treatment). **F** Double IF staining of CD20, CD8 and DAPI in triple combination mice (Representative image of $n = 5$ mice). **G** Quantification MFI staining of CSF-1R, BrdU, CD20 and CD8 following treatments.

Significance was determined by Two-way ANOVA with multiple comparisons test ($n = 6$ independent ROI). **H** Tumor volume of control (naïve) mice and rechallenged mice. Significance determined by Two-way ANOVA and Šidák's multiple comparisons test, ($n = 5$ control mice and $n = 6$ CR mice). **I** Schematic illustration of ACT. **J** Tumor volume of mice bearing T12 tumors subjected to ACT treated with IgG control and +/-aPD-1. Significance was determined by Two-way ANOVA with multiple comparisons test, ($n = 5$ IgG, $n = 5$ aPD-1, $n = 7$ T cells and $n = 8$ aPD-1 + T cells). **K** Representative images of isolated spleens (image is representative of $n = 3$ mice per treatment). **L** Quantification of IHC staining of CD8 T-cells. Significance determined by ordinary one-way ANOVA, ($n = 8$ independent ROI). **M** Quantification of IHC staining of CD8 T-cells of ACT transferred mice. Significance determined by ordinary one-way ANOVA, ($n = 8$ independent ROI). Data shown as mean ± SEM.

### Tail vein (TV) injection and portal vein (PV) injection

For experimental metastases, freshly digested tumor cells were prepared from T12 tumor tissues. T12 tumors were transplanted into the mammary gland of a WT BALB/c mouse as described above. When the tumor reached ~1 cm in diameter, the tumor was harvested and processed into a single-cell suspension. Briefly, tumors were digested with 1 mg/mL collagenase type I (Sigma-Aldrich, 11088793001) and 1 µg/mL DNase (Sigma-Aldrich, 11284932001) for 2 h at 37 °C on a shaker. Three rounds of rapid centrifugation were performed to enrich tumor cells by isolating stromal cells. The cell pellets were then trypsinized, counted, and filtered into single cell suspension. Established lung metastases were generated via TV injection of 30,000 unlabeled freshly dissociated cells suspended in 200 µL of PBS into each 8- to 10 week-old female BALB/c mice. PV injections were performed following established protocols[58]. Briefly, after anesthetizing the animals and sterilizing the surgical site, a small incision (1-inch) was made between the median and sagittal planes on the left side of the mouse. 3000 (unlabeled) freshly digested tumor cells, suspended in 5 µL of PBS were then slowly injected into the portal vein using a 32-gauge needle (Hamilton). A hemostatic gauze with gentle pressure will be placed at the injection site to manage the bleeding before closing the wound. The ethical endpoint was met when the mice lost ~20% body weight or if the mice exhibited ruffled fur and or hunched posture. To ensure successful generation of lung or liver metastases, several mice were injected with 60 mg/kg BrdU (Sigma-Aldrich, B-5002) 2 h prior to sacrifice to identify proliferative cells. Mice that experience lung and or liver metastases rapidly decline, thus body weight was measured every 3 to 7 days. The ethical endpoint was met when mice either lost ~20% body weight or display poor body condition.

### In vivo treatments

Anti-CSF-1R monoclonal antibody, SNDX-ms6352 was donated by Syndax Pharmaceuticals, Inc. and was administered via i.p. injection weekly for 4 weeks. Per Syndax recommendation, mice were administered an initial dose of 40 mg/kg was first administered, followed by 20 mg/kg for the remaining three doses. Control mice were injected with equal volume of InVivoMab mouse IgG1 isotype control (Bio X Cell, BE0083). Cyclophosphamide (Sigma-Aldrich, PHR1404-1G) was resuspended in sterile PBS (Lonza). Mice were injected i.p. at the concentration of 100 mg/kg, once a week for up to 4 weeks. Control mice were injected with the same volume of sterile PBS. For ICB treatment, mice were administered RecombiMab anti-mouse PD1 (BioXcell, CD279, CP151) and RecombiMAb anti-mouse IgG2a isotype (BioXcell, CP150) 200 µg per mouse every 3 days via i.p. injection. In vivo antibodies were diluted with InVivoPure pH 7.0 Dilution Buffer (BioXCell, IP0070).

### Tumor associated macrophage isolation and in vitro treatments

TAMs were directly isolated from T12 primary mammary tumors transplanted in female BALB/c mice using mouse CD11b positive selection beads (STMECELL Technologies, 18970). Briefly primary tumors were harvested and processed into a single-cell suspension as previously described. Following quick centrifugation, the stromal compartment (macrophage enriched) was utilized. Stromal cells were resuspended in DMEM/F12 medium (Thermo Fisher Scientific, 11330032) with 2% FBS. Stromal cells were then filtered, counted and isolated using the EasySep Magnet (STEMCELL Technologies, 18000). TAMs were then cultured in the appropriate TAM medium consisting of RPMI-1640, no glutamine (Thermo Fisher Scientific, 21870076) 20% FBS, 200 µM L-glutamine (Thermo Fisher Scientific, 25030081), 100 µM sodium pyruvate (Thermo Fisher Scientific, 11360070), 55 µM 2-mercaptoethanol (Thermo Fisher Scientific, 21-985-023), 10 ng/ml mouse M-CSF (Biolegend, 576406) and MEM Non-Essential Amino Acids Solution (Thermo Fisher Scientific, 11140050). Non-adherent cells were washed away post 2 h incubation with PBS. Subsequently, 500,000 TAMs were cultured in 6 well-plates and treated with increasing concentration of SNDX-ms6352 (Syndax Pharmaceuticals, Inc.) or IgG1 isotype control (Bio X Cell, BE0083). TAM growth was monitored using the IncuCyte S3 Live-Cell Analysis System.

### Immunoblotting

For immunoblotting, TAMs were detached from the culture dish and lysed using RIPA buffer (10 mM Tris-HCl, pH 8.0, 1 mM EDTA, 0.5 mM EGTA, 1% Triton X100, 0.1% Sodium Deoxycholate, 0.1% SDS, 140 mM NaCl). BCA assay (Pierce BCA Protein Assay Kit, The rmo Fisher Scientific, 23225) was used to determine protein concentration. Equal amount of protein was separated by SDS-PAGE using Mini-Protean TGX polyacrylamide gels (Bio-Rad). Subsequently, proteins were transferred from the gel to the Immobilon PVDF membranes (Millipore, IPVH00010). Following a 1 h blocking step with 5% BSA, the membranes were incubated with anti-CSF-1R (Cell Signaling TECHNOLOGY, 43390) and anti-CD206/MRC1 (Cell Signaling Technology, 24595) overnight at 4°C with primary antibodies. The following day, membranes were probed with secondary HRP antibodies Goat anti-Rabbit IgG (Invitrogen, 31460) and scanned using the Amersham Imager 600 system.

### Immunohistochemistry and immunofluorescence

Primary tumors, mammary glands, lung and liver tissues were fixed in 4% paraformaldehyde for 24 h at 4°C and subsequently changed to 70% EtOH prior to paraffinization and embedding. For lung and liver metastases, 5-6 serial sections (5-µm per section) were collected every 150 µm.

For H&E staining, tissues were deparaffinized and rehydrated and stained Harris Hematoxylin (Poly Scientific, S176). For IHC analysis sections were first deparaffinized and rehydrated, antigen retrieval was conducted using citrate buffer (pH 6.0) placed on a steamer for 30 min. Endogenous peroxidase was performed using 3% $H_2O_2$ (Thermo Fisher Scientific, H323-500) in PBS for 10 min. Sections were then incubated in the appropriate blocking buffer containing 3% BSA

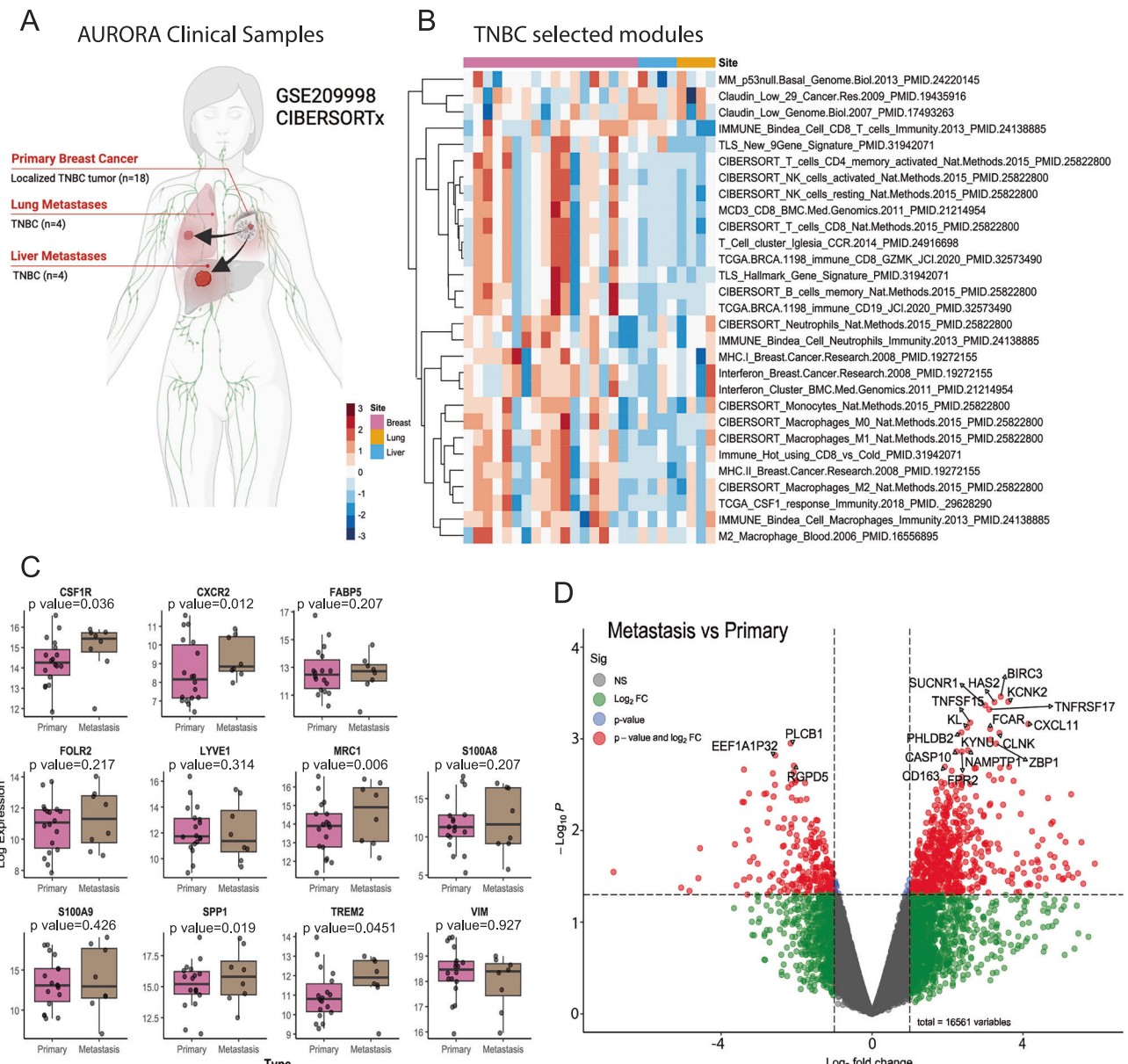

**Fig. 7 | Patients display elevated immunosuppressive macrophage signatures in TNBC lung and liver metastatic sites. A** Schematic illustration of AURORA US clinical samples from TNBC primary, lung and liver metastatic sites. (*n = 18* primary, *n = 4* lung and *n = 4* liver). **B** Representative heatmap of various immunological and claudin-low/p53 signatures between primary and metastatic tumors from TNBC patients (*n = 18* primary, *n = 4* lung and *n = 4* liver). **C** Gene expression levels of macrophage, neutrophil and EMT signatures between TNBC primary and metastatic tumors (*n = 18* primary versus *n = 8* metastatic tumors). Box plots display the median (center line), interquartile (25th -75th percentiles) and whiskers extending to the minimum and maximum values, outliers are shown as individual points. **D** Volcano plot depicting the most up- and downregulated genes in metastatic samples compared to primary tumors. For all multiple comparisons, *P*-values were adjusted using the Benjamini–Hochberg method to control the false discovery rate (FDR). Gene set enrichment analyses were performed using the fgsea package in R. Gene sets with adjusted *P*-values < 0.05 were considered statistically significant.

(Sigma-Aldrich, A7906) and 5% goat serum (Sigma-Aldrich, G9023) in PBS for 1 h at room temperature and incubated in the respective primary antibodies overnight at 4°C. For IHC the following primary antibodies were utilized: anti-F4/80 (Cell Signaling Technology, 70076; 1:500), anti-S100A8 (R&D Systems, MAB3059; 1:5000), anti-CD8α (Cell Signaling Technology, 98941; 1:500) and anti-CD4 (Abcam, ab183685 1:400). Biotin-conjugated secondary antibodies, anti-rabbit (Vector Laboratories, PI-1000-1) or anti-rat (Vector Laboratories, PI-9400-1) were incubated for 1 h at room temperature. VECTASTAIN Elite ABC HRP Reagent (Vector Laboratories, PK7100) was used to amplify signal according to the manufacture's protocol and developed with ImmPACT DAB peroxidase substrate (Vector Laboratories, sk-4105).

The slides were counterstained with Haris Hematoxylin (Poly Scientific, S212A). IHC and H&E-stained slides were imaged using the Olympus BX40 light microscope and MPX-5C pro low-light camera at x20 magnification or scanned with the Aperio ImageScope (Leica Biosystems) and analyzed using the Aperio ImageScope software (v12.3.3.5048). At least three representative images of primary tumor, lung and liver metastases were acquired for analysis.

For immunofluorescence analysis deparaffinized and rehydrated slides were subjected to antigen retrieval using Tris-EDTA antigen retrieval buffer (10 mM Tris, 1 mM EDTA, 0.05% Tween 20, pH 9.0). The following primary antibodies were incubated overnight at 4°C: anti-F4/80 (Cell Signaling Technology, 70076; 1:500), anti-S100A8 (R&D

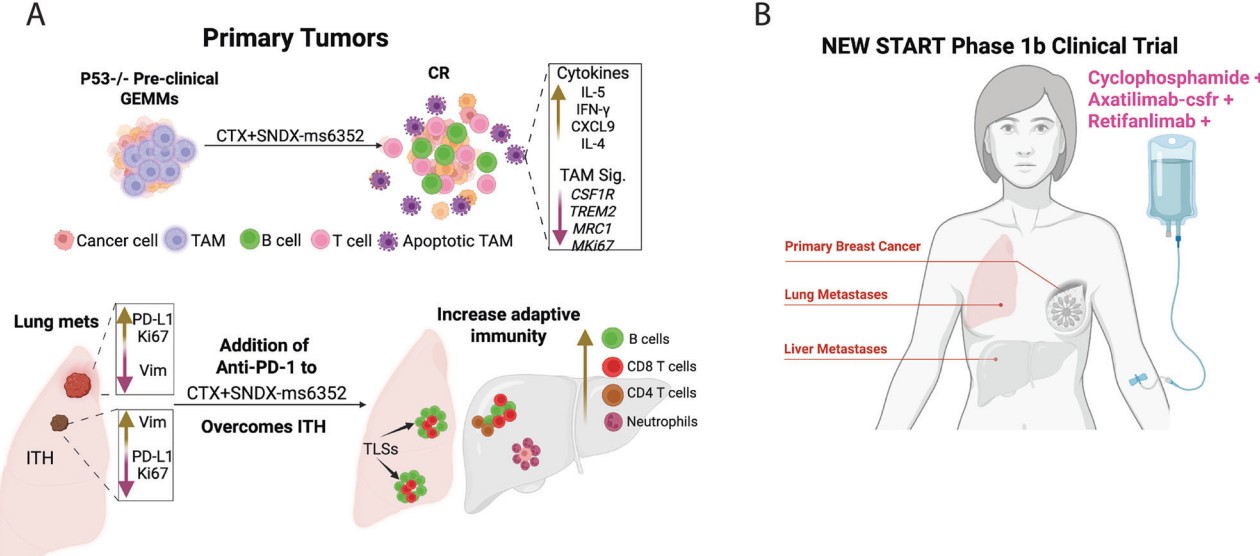

**Fig. 8 | Triple combination promotes adaptive immunity in pre-clinical models and will now move to a phase 1b clinical trial. A** Ablating TAMs with SNDX-ms6352 and immunostimulatory CTX upregulated anti-tumor cytokines (IL-5, IFN-y, CXCL9 and IL-4) while suppressing TAM signatures (*CSF1R, TREM2 and MRC1*) to prolong survival and obtain a CR in primary syngeneic model. The addition of anti-PD-1 is required to overcome ITH associated recurrence in double combination treated syngeneic model harboring lung metastases. Triple combination treated syngeneic model displayed TLSs in the lung while liver metastases had elevated CD8[+], CD4[+] T cells and CD20[+] B cells with neutrophils surrounding residual tumor cells. **B** The combination of CTX with Axatilimab-csfr (SNDX-ms6352) and Retifanlimab (anti-PD-1) will now be tested as a phase 1b clinical study in TNBC patients with lung and liver metastases.

Systems, MAB3059; 1:5000), anti-BrdU (1:1000; Abcam, ab6326), anti−cleaved caspase 3 (Cell Signaling Technology, 9661, 1:1000), anti-CD8α (Cell Signaling Technology, 98941; 1:500), anti-CD20 (Cell Signaling Technology, 70168; 1:500), anti-CSF-1R/M-CSF-R (Cell Signaling Technology, 43390, 1:500) and anti-CD8α (eBioscience, 14-0081-82; 1:500). For immunofluorescence staining of actin and F4/80 in TAMs, the isolated TAMs were first cultured in 8-well chamber cell culture slides (Corning, 354108) 48 h after treatment, fixed in 4% paraformaldehyde for 15 min and permeabilized with 0.5% Triton X-100 followed by overnight blocking in 3% BSA at 4 °C. Following several washes with PBS both slide and cells were incubated in the appropriate conjugated secondary antibody (Alexa Fluor 488/594/647) for 1 h at room temperature. The slides were counterstained with DAPI (Invitrogen™, R37606) for 20 min at room temperature and mounted using aqua-Poly/Mount (Fisher Scientific, NC9439247). Immunofluorescent images were taken using the Nikon A1-Rs confocal microscope at x20 and x40 magnification and quantified using the Fiji software.

## Cytokine profiling
Snap-frozen tumor tissues from in vivo treatments were thawed and homogenized using zirconium beads in the BeadBlaster 24 Microtube Homogenizer. The tissues were homogenized in the T-PER Tissue Protein Extraction Reagent (Thermo Fisher Scientific, 78510) with EDTA-free Protease Inhibitor Cocktail. The BCA Protein Assay Kit was utilized to determine protein concentrations. Equal amount of homogenate was measured and diluted 2-fold in PBS. Samples were profiled using the Mouse Cytokine/Chemokine 44-Plex Discovery Assay Array (Eve Technologies Corp., MD44). All cytokine levels were normalized to IgG Controls.

## Imaging mass cytometry (IMC)
Formalin-fixed paraffin-embedded tissues were deparaffinized in fresh 100% Xyelene three times for 5 min and rehydrated in a series of ethanol (100%, 100%, 95%, 80% 70%) 5 min each. Washed with deionized water for 5 min with gentle agitation. Antigen retrieval was performed using Tris-EDTA (pH 9.0) (eBioscience, 00-4956-58), incubated

at 95 °C for 30 min in a steamer and allowed to cool in the same solution for 30 min. The slides were then washed with deionized water for 10 min and subsequently blocked with 3% BSA in PBS for 1 h at room temperature in a hydration chamber. The slides were then incubated with the respective metal tagged antibody cocktail overnight at 4 °C in a hydration chamber. The slides were then washed 2 times in PBS-T for 5 min and 2 more times in PBS prior to being incubated with 0.125 μm of Interculator-Ir (Standard Biotools) in PBS for 30 min at room temperature in a hydration chamber. The slides were then washed in deionized water for 5 min and left to air dry for 20 min at room temperature.

Stained and dried tissue slides were loaded into the Hyperion tissue imager and ablated by a high energy UV laser beam 1mm2/pixel at 200 Hz. The ablated material is transported by argon gas into the plasma of the mass cytometer, where first atomized/ionized, then low-molecules ions filtered and analyzed in the time-of-flight (TOF) mass spectrometer. Multiplexing Image data with spatial information are reconstructed and saved in MCD. Format for down-stream analysis. Cell-level image quality-control is first evaluated with the MCD viewer software provided by Standard Biotools. The full resolution IMC images (10X) in MCD. format is added to the Visiopharm® database and color-adjusted for best visualization for nuclear detection and segmentation. Cell segmentation is performed using DNA signal to find the nuclei followed by Visiopharm® designated algorithms including polynomial blobs to refine the separation of cells. Cellular phenotyping is then performed, and spatial data (Nearest Neighbor) calculated using Visiopharm® designated Phenoplex Guided Workflow. Analyzed data with all cells and their phenotypes were exported to TSV. file for further analysis. TSV files were first processed in R using FlowSOM, flowCore and SingleCellExperiment packages in preparation for further analyses using the CATALYST package (version 1.14.0, https://github.com/HelenaLC/CATALYST). The markers included in the analyses were Pan-CK, CD44, F4/80, S100A8, S100A9, PD-L1, Ki67, CD4, CD8a, Vimentin and PD-1. Clustering was performed using the cluster (CATALYST) function (maxK = 20). Cluster annotations were performed manually (neutrophils: S100A8/S100A9; macrophages: F4/80;

tumor cells: Vimentin/CD44). The clusters which lacked the expression of these markers were labeled as unidentified. Dimension reduction by TSNE was performed and plotted using runDR (CATALYST) function using 5000 cells per sample. Clusters were merged based on annotations using the mergeClusters (CATALYST) function. All plots were generated using the CATALYST package.

For cluster analysis IMC markers were divided into those that are expressed in the nucleus, such as DNA (channels 191Ir and 193Ir), Ki67, Ly6G, B220, F480, S100A8/9, and those that are expressed in the cell membrane, such as CD4, CD8a, CD11b, CD11c, CD44, CD62L, CD68. Composite images were generated for nucleus and cell membrane. Next, a combined image of the cells (nucleus + cell membrane) was generated for the purpose of cell segmentations. To perform cell segmentation, we first converted each image to grayscale, then highlighted in-cell pixels using the Threshold menu option in ImageJ. Approximately 40–55% of pixels per image were highlighted as red (i.e., in-cell pixels). With these in-cell pixels highlighted as red on the screen, we next used "Find Maxima" function, followed by setting output to "Segmented Particles" in ImageJ to create black segmented cell masks. Results were outputted to the ROI manager and subsequently saved as a ROISet.zip file. We next quantified the total intensities of pixels per protein marker per cell (given the available cell masks created from segmentation). Eventually, this created a cell-by-protein matrix for all segmented single-cells and protein markers in each image. On this matrix, we performed log-transformation, followed by row-wise z-scoring (across cells), and column-wise z-scoring (across markers). The resulting matrix was subject to K-means clustering with a high number of random starts (nstart = 100,000) for reliable centroid generation was used. K-means promoted greater tolerance for variation and noise in data processing, were less sensitive and more robust to outlier expression, and avoided the generation of ROI -specific clusters. 20 clusters were determined showing robust cluster-specific expression. Differentially expressed proteins across clusters were then determined by performing all one-vs-one comparisons and then setting a threshold for significance. This threshold was set at 17/19, meaning that a protein is deemed to be differentially expressed for a cluster if the protein is significant in 17 out of 19 one-vs-one comparisons between that cluster and the rest. After that, each cluster was named by the list of differential markers expressed in the cluster.

Image acquisition was performed using the available histoCAT web (http://github.com/BodenmillerGroup/histocat-web) software. The antibody panel was designed to study the TIME within metastatic lesions; thus we included markers to identify tumor cells (PanCK, Vimentin, CD44), proliferation marker (Ki-67), lymphocytes (CD8, CD4, FoxP3, CD127, CD62L, B220), Monocytes (F4/80, S100A8, S100A9, Ly6C, Ly6G, CD163, CD11c, CD86) and the checkpoint markers (PD-L1, PD-1). Multiple iridium staining makers (191Ir, 193Ir) were used for nuclei detection. The antibody panel was designed from our previous study[15] and literature search.

For antibody validation a separate tissue microarray (TMA) was generated using normal spleen, normal lung, lung metastases and primary tumor tissues from paraffin embedded blocks, sectioned at 5 μm. Each specific region of the slide contained the four varying tissues and was separated using a hydrophobic PAP pen. The TMA was generated by the Human Tissue Acquisition and Pathology Core and inspected by a pathologist to assess tissues. Four different antibody cocktail dilutions were then tested; each cocktail contained 23 different antibodies. Multiple antigen retrieval buffers were simultaneously tested with each cocktail dilution, including citrate buffer (pH 6.0) and IHC Antigen Retrieval Solution (10 mM Tris, 1 mM EDTA, pH 9.0) (eBioscience, 00-4956-58). The different cocktail groups were obtained analyzed and validated in conjunction with The Cytometry and Cell Sorting Core (CCSC) at Baylor College of Medicine (BCM).

## Flow cytometry

For the enrichment of stromal cells, single cell suspensions were generated following the digestion of tumor tissues with collagenase as previously described. Briefly, stromal cells were collected following quick centrifugations and incubated in RBC lysis buffer (BioLegend, 420301), the cells were resuspended following lysis inactivation, counted and filtered PBS + 2% FBS. Prior to blocking, anti-mouse CD16/CD32 antibody (BioLegend, 101320) 1 million cells were stained with Live/Dead Fixable Yellow (Invitrogen™, L34968; 1:800). Stromal cells were incubated with the cell surface F4/80-APC (BioLegend, 123115). Following antibody staining, the resuspended cells (PBS) were analyzed using the Attune NxT flow cytometer at the FACS and CCSC at BCM. Compensation and further analyses were performed using the FlowJo v10 software.

## Single-cell RNA-sequencing (scRNA-seq)

For scRNA-seq, all mice received the final doses 24 h prior to euthanasia. Single cell suspensions were processed as previously described and pooled from 3 mice per treatment. Only viable cells were used for single cell analysis as such, isolated single cell suspensions were first stained with Ghost Dye UV 450 (Tonbo Biosciences, 13-0868) for 10 min and sorted via FACS to isolate viable cells. Briefly, primary tumor cells were harvested and resuspended in RBC Lysis Buffer (BioLegend, Inc., 420301) and passed through a 70- μm strainer. Viable single cell suspensions were tagged with 3' CellPlex barcoding oligos (10X Genomics, PN-1000262) and were quickly subjected for scRNA-seq library preparation by the Single Cell Genomics Core at BCM. Single-cell libraries were prepared following the Chromium Single Cell Gene Expression 3' v3.1 kit encompassing the multiplex capture CMO barcoding kit (10x Genomics, PN-1000262). Next generation library sequencing was performed using the NovaSeq 6000 (Illumina) following quality control check.

Using the raw sequencing data, alignment, read counts and sample demultiplexing was performed using the CellRanger multi (v7.2.0) pipeline. The barcode assignment confidence was set to 0.8 to capture a larger number of cells for downstream processing. scRNA-seq was analyzed using the Seurat (v4.4.0) package in R (R version 4.3.1; https://github.com/satijalab/seurat). Only cells with <6000 and >200 read counts were utilized for downstream analysis to accommodate for quality control. Cells with >10% mitochondrial ratio and over 1000 UMI were removed. Individual datasets were batch-corrected and integrated using the IntegrateData function. The circle plot was generated by CellChat (netVisual_circle function).

## Adoptive T cell transfer

T cells were directly isolated from the spleens of female BALB/c mice that were bearing T12 liver metastases and had been previously treated for 56 days and rejected tumor cell rechallenge. Briefly, spleens were first dissociated using the Spleen Dissociation Medium (STEMCELL Technologies, 07915). Splenocytes were isolated using the EasySEP Mouse T Cell Isolation Kit (STMECELL Technologies, 19851), negative selection was performed using the EasySep Magnet (STEMCELL Technologies, 18000). T cells were then resuspended in PBS, counted and directly transferred into T12 primary tumor bearing female BALB/c via TV injection and treated with RecombiMab anti-mouse PD1 (BioX-cell, CD279, CP151) or RecombiMAb anti-mouse IgG2a isotype (BioX-cell, CP150) 200 μg per mouse.

## Radiopharmaceuticals (FDG) and Small-Animal PET-CT

Fluorine-18 labeled fluoroxyglucose (18F-FDG), purchased from Sofie, (Houston, TX). All CT and PET images were acquired using an Inveon scanner (Siemens AG, Knoxville, TN). The mice were injected with 11.1 MBq (300 μCi) of 18F-FDG radiotracer at any given time. To identify metastases or measure tumor metabolic activities, 18F-FDG injected intra peritoneal. Before 18F-FDG

administration, the mice were fasted for ~12 h. PET and CT performed 1 h after injection of FDG. During imaging, a respiratory pad was placed under the abdomen of the animal to monitor respiration (Biovet, Newark, NJ). Mice were anesthetized with isoflurane gas (1–3%) mixed with oxygen at a flow rate of 0.5-1 L/min and adjusted accordingly during imaging to maintain normal breathing rates. A CT scan acquired with the following specifications: 220 acquired projections each projection was 290 ms with x-ray tube voltage and current set at 60 kVp and 500 µA, respectively. A 40 min PET scan immediately acquired after CT. The PET scans were reconstructed using OSEM3D reconstruction method and co-registered to the CT scan for attenuation correction.

## PET image analysis

The PET images were quantified using Inveon Research Workspace IRW (IRW, Siemens AG, Knoxville, TN). Using the reconstructed PET scan, lung and liver manually selected to form regions of interest (ROI) on the PET-CT images. Activity measurements (Bq/cm3) were divided by the decay-corrected injected dose (Bq) and multiplied by 100 to calculate tissue uptake index represented as percentage injected dose per gram of tissue. The data represented as standardized uptake value (SUV) normalized to body weight.

## AURORA data analysis

The batch-corrected and normal tissue-adjusted gene expression and expression signature data were obtained from the AURORA USA clinical study (GSE209998)[11]. For primary figures the data were filtered to contain only triple-negative primary tumors, lung metastases, and liver metastases (Supplementary Table 3 and Dataset S14). Differential gene expression/expression signature analyses were performed using the limma package. Patient IDs were treated as a corresponding variable. For all multiple comparisons, $P$-values were adjusted using the Benjamini–Hochberg method to control the false discovery rate (FDR). Gene set enrichment analyses were performed using the fgsea package in R. Gene sets with adjusted $P$-values <0.05 were considered statistically significant.

## Statistics

Detailed statistical methods are outlined within the figure legends of this manuscript, depicting the number of animals or independent replicates, along with the statistical tests used. For pairwise comparisons or comparisons of 3 or more groups ordinary one-way ANOVA was used. Two-way ANOVA and Šidák's multiple-comparison test were used for analyzing tumor volume changes, log (fold changes) of tumor volume and Tukey's multiple-comparison test were used for body weight changes over time. For Kaplan-Meier curves, the log-rank test was performed. For comparison between 2 groups, unpaired, 2-tailed Student's $t$-test was used. Paired $t$-test was used for PET/CT analysis. The above-mentioned statistics were calculated using GraphPad Prism Version 10.3.0. For scRNA-seq, statistical analyses were performed in R. Log2 (fold change) >0.5 or <−0.05, and an adjusted $P$-value of <0.01 was considered significant in differential gene expression analyses (Supplementary Datasets S1–S11). An adjusted $P$-value of <0.05 was considered significant in GO pathway enrichment analyses. For IMC, the statistical analyses of differential expression were performed using the multcomp and diffcyt packages with all clusters merged. Contrast matrices were created for each comparison between two conditions. $P$-values were generated using diffcyt function (type of analysis, DS; method for DS testing, diffcyt-DS-limma). $P$-values were then adjusted using the Benjamini-Hochberg method.

## Reporting summary

Further information on research design is available in the Nature Portfolio Reporting Summary linked to this article.

## Data availability

Raw and analyzed scRNA-seq data has been deposited in the NCBI GEO database accession number (GSE292908). Detailed information regarding the antibodies used for IMC is provided in Supplementary table 4, IMC single cell analysis provided in Supplementary dataset S15-17. All unique/stable reagents generated in this study are available from the Lead Contact following a completed Materials Transfer Agreement (MTA). Source data are provided with this paper. All illustrations were created with BioRender. Further information and requests for resources and reagents should be directed to and will be fulfilled by the Lead Contact, Jeffrey M. Rosen (jrosen@bcm.edu).

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

## Acknowledgements

We are grateful for the suggestions from both the Zhang and Rosen laboratory members. We thank Peter Ordentlich from Syndax for providing SNDX-ms6352 and Dr. Patricia Castro from the Human Tissue Acquisition and Pathology Core at BCM for IMC antibody validation. D.A.P. is supported by the American Cancer Society Postdoctoral Fellowship (PF-22-163-01-MM) and NIGMS MOSAIC (K99GM155594). A.J.S. is supported by Susan G Komen (SAC232150). N.Z. is supported by Cancer Prevention and Research Institute of Texas (CPRIT) (RP220468). C.M.P. was supported by funds from NCI Breast SPORE program (P50-CA058223) and U01-CA289839-01. X.H.-F.Z. is supported by US Department of Defense (DAMD W81XWH-16-1-0073 and DAMD W81XWH-20-1-0375), NIH (R01CA183878, R01CA227904, R01CA221946, R01CA251950 and 5P50CA186784-03) the Breast Cancer Research Foundation, and McNair Foundation. J.M.R. was supported by NIH (R01CA016303-46 and R01CA148761-13) and Susan G. Komen (SAC232150). This project was supported by the Cytometry and Cell Sorting Core at Baylor College of Medicine with funding from the CPRIT Core Facility Support Award (CPRIT-RP180672), the NIH (CA125123 and RR024574) and the assistance of Joel M. Sederstrom. The scRNA-seq experiments were supported by the Single Cell Genomics Core at BCM, which is partially supported by NIH (S10OD025240) and CPRIT

(RP200504). The Integrated Microscopy core supported by NIH (DK56338, CA125123, ES030285), and CPRIT (RP150578, RP170719). We would like to thank Texas Children's Hospital for the use of the Small Animal Imaging facility (SAIF) for getting PET imaging. We thank the Pathology Core of Lester and Sue Smith Breast Center at BCM for tissue sectioning. The content is solely the responsibility of the authors and does not necessarily represent the official views of the National Institutes of Health.

## Author contributions

D.A.P. conceptualized and designed the study, conducted the experiments, analyzed the data, and wrote the manuscript. X.Y., F.L., H.L.C., W.B. and Q.Z. provided bioinformatics support. C.Z., S.J.C., N.L., C.V.O., O.P. conducted experiments and analyzed data. W.W and P.P analyzed imaging mass cytometry experiments. A.J.S. and N.Z. designed the flow cytometry experiments and analyzed the data. P.S. and M.W.G. conducted PET/CT scans and analyzed the data. C.M.P. provided bioinformatics support and edited the manuscript. X.H.-F.Z. and J.M.R. conceived and supervised the study and edited the manuscript.

## Competing interests

C.M.P. is an equity stockholder of and consultant for BioClassifier LLC. C.M.P. is also listed as an inventor on a patent application (US9631239B2) for the Breast PAM50 Subtyping assay (not related to this work). The remaining authors declare no competing interests.

## Additional information

[1]Department of Molecular and Cellular Biology, Baylor College of Medicine, One Baylor Plaza, Houston, TX 77030, USA. [2]Lester and Sue Smith Breast Center, Baylor College of Medicine, One Baylor Plaza, Houston, TX 77030, USA. [3]Dan L. Duncan Comprehensive Cancer Center, Baylor College of Medicine, One Baylor Plaza, Houston, TX 77030, USA. [4]Graduate Program in Cancer and Cell Biology, Baylor College of Medicine, One Baylor Plaza, Houston, TX 77030, USA. [5]Medical Scientist Training Program, Baylor College of Medicine, Houston, TX 77030, USA. [6]Department of Molecular and Human Genetics, Baylor College of Medicine, One Baylor Plaza, Houston, TX 77030, USA. [7]Cytometry and Cell Sorting Core, Baylor College of Medicine, One Baylor Plaza, Houston, TX 77030, USA. [8]Department of Pediatrics, Baylor College of Medicine, One Baylor Plaza, Houston, TX 77030, USA. [9]Lineberger Comprehensive Cancer Center, University of North Carolina, 450 West Dr, Chapel Hill, NC 27599, USA. [10]McNair Medical Institute, Baylor College of Medicine, One Baylor Plaza, Houston, TX 77030, USA. ✉e-mail: jrosen@bcm.edu

