## [Transparent Peer Review file · Nature Communications]

Anti-CSF-1R therapy with combined immuno- chemotherapy coordinate an adaptive immune response to eliminate macrophage enriched Triple Negative Breast Cancers.

Corresponding Author: Professor Jeffrey Rosen

Version 0:

Reviewer comments:

Reviewer #1

(Remarks to the Author)

In the manuscript by Pedroza et al, the authors developed a new combination treatment for TNBC which is now entering the first phase of a clinical trial. They successfully demonstrated the ability of this treatment to increase survival in a TNBC model. In addition, they tested its effect on lung and liver metastatic models confirming its efficacy.

The manuscript is well written, and it introduces a new treatment for TNBC. However, there are some major issues that need to be addressed and the mechanism behind the therapy efficacy is lacking.

Major comments:

- The rationale for using axatilimab-csfr instead of the frequently used PLX-3397 was to reduce off target effects as PLX-3397 is accompanied with toxicity. However, except for monitoring the tumor weight, in this study there are no analysis related to the toxicity of the anti-CSF1R compound and the combination with CTX. This small inhibitor is reported to induce liver toxicity, so it is important to prove that the mAb does not result in similar adverse effects as the ones induced by PLX-3397.
- In the first experiment the authors tested the combination treatment in five different models and observed a strong increase in survival in two of them. In the remaining part of the study, they use mostly only one of these responsive models. It is critical to understand why the other models had only a partial, as these could represent non- or less-responsive patients. In addition, they only show the depletion effect of the anti-csf1r mAb on the T12 model (Fig S1D). Is the therapy not as potent in the other models because the Mf are not well depleted? Or because the niche is filled by other immunosuppressive cells such as neutrophils? Understanding the resistance mechanism and proposing a combination therapy for the unresponsive models would strongly improve the impact of the manuscript.
- In the first paragraph the authors reported to have treated "several Trp53 null TNBC GEMMs..". First, the authors claim to use GEMM models, however these are transplantable tumors which are not GEMMs. Secondly, "several" is a vague term, they use five models, of which three are claudin low, one is basal like and one luminal like. The text should be more precise and clear for the reader. Especially because the authors claim to have a treatment for TNBC, but it seems that this treatment would be efficient only in a subset of patients with a claudin-low trp53 null tumor.
- The manuscript heavily relies on the two "metastatic models" obtained with tail vein injection for the lungs and portal vein injection for the liver. Although these are easy to obtained models, as the authors highlight in the discussion, these models of experimental metastasis do not reflect the properties of spontaneous metastasis. Hence, some of the results should be recapitulated in a spontaneous metastatic model after resection. This model would reflect better the reality of patient treatment. Moreover, it is not clear why mice with lung metastases could not survive the PET/CT scan, this is frequently done.
- A major point is that in general, the mechanism behind the efficiency of the treatment is not clear. The authors make several observations regarding cell infiltration, cytokines levels, but no real mechanism is put forth. Are the remaining Mf repolarized in a way they induce a Th1 response, leading to IFN γ accumulation, subsequently leading to PDL1 upregulation? Is the therapeutic effect fully T-cell mediated? What is the role of B cells and NK cells in the therapeutic efficacy? What is the role of

IL-17 and IL-5 and by who are these produced. Is the cytokine production linked to CSF1R signaling as mentioned in the abstract? At least some mechanistical insights behind the mode of action should be provided.

- Finally, is the addition of anti-PD1 treatment really necessary? In Fig. 2C we can observe that approximately 70% of the mice treated with the combination are alive at 56 days. In fig. 4B mice treated with the same combination + anti-PD1 are all dead at this time point. Can the authors clarify on this? In addition, the mice receiving the m. dose of PD1 were only monitored until 70 days. Did the surviving mice showed sign of metastasis? Also in the case of liver metastasis, Fig. 6D, the mice were only monitored until day 56, these mice could still developed metastasis. After the rechallenge, did the authors check the livers of these mice?

Minor comments:

In general the figures are quite blurry and difficult to read. This might be due to the compression during the submission, but it would be good to have high quality figures in the revised version.

- Fig 1F and S1J: the authors claim that these two models are resistant to tumor rechallenge. However, we can still observe tumor growth, which appears to be just delayed, so the models can't be really considered "resistant". The authors should rephrase their claims.

- Some typos, such as in the abstract: IL-15 instead of IL-5, Several times INF is written instead of IFN

- Fig 1J, how was this circle plot analysis performed?

- Fig 1G, in the umap some names are covering each-other

- Fig 1K, the plasma from IgG control mice is missing from this analysis.

- Fig. 2F and S3D: what are con1&2 and Combos 1&2? Were only 2 lungs used for the analysis? Fig. S3E-F: what is considered a MaMet and what a MiMet? In general, for all the IMC experiments is not clear how many samples were analyzed.

- What is the difference between 2B and S2A?

- Page 10, referring to fig. 4F the authors talk about double combination, while the staining has been done on the triple combination.

- Fig. 6C, in the legend is reported that the analysis has been done on 3 mice per group, but on the figure there are represented only 2 dots.

Reviewer #2

(Remarks to the Author)

This group builds on previous data published in Cancer Research (PMID: 35442423) to investigate the targeted inhibition of CSF-1R in combination with cyclophosphamide (CTX), along with anti-PD1 in experimental lung and liver metastasis using their transplantable GEM models that recapitulate mutant-p53 claudin-low subtypes. The previous paper did include analysis of the dual combination on orthotopic tumor growth (in a couple of GEM-derived models)– yet this paper brings in metastasis and addition of anti-PD1. Targeting CSF-R1 for breast cancer metastasis, alone or in combination with chemotherapy or PD-1 antibodies, is not novel, However, the use of the more specific SNDX-ms6352 antibody and the mechanisms at play in the TME are important to guide future clinical trials.

Main points that reduce enthusiasm for publication is the lack of metastasis models used, which is important when using experimental rather than spontaneous models. The authors focus the metastasis studies on one model- the T12 model. Given that the addition of anti-PD1 had most benefit in the liver metastasis model (more-so than lung), this should be expanded to determine the clinical relevance for publication in Nature Comms.

Specific comments/questions are:

1) Figure 1D- only the highly neutrophil enriched luminal-like 2208L model failed to respond to the combination. Given this model still has more macrophages than neutrophils, what is going on here? Need to block both? It would have been interesting to include this model in the triple combination to enable the link between response and the state of the TME pre- and post-therapy.

2) Figure 1G- an impressive difference in T an B cells and decrease in macrophages with combination- but was this the only replicate for each treatment?

3) Figure 1J- is this really the best way to depict this? It's confusing.

4) Figure 1K- as far as I can see, the no treatment control is missing here and that should be included to determine if any single agent is detrimental or the opposite trend is observed.

5) Figure 2C- "only the combination led to better survival even after treatment cessation" this is not correct as chemotherapy alone did as well. This is critical given the novel angle of this paper and that, in figure 4, the addition of anti-PD1 with CTX is no different to the triple combination.

6) Figure 2F/G- Although Combos 1&2 expressed decreased proliferative tumor cells (Ki67+, Vim+, CD44+) these immunologically cold micro-metastases displayed elevated levels of

exhaustion markers PD-L1 and PD-1 - from the data presented this is incorrect and only applies to PD-1 as the tumour cells in the control already had high PD-L1. Why was only combination of controls? Single agents should be looked at as well.

7) Page 9- PD-L1 is a well-known T cell exhaustion marker that is not commonly observed in non- metastatic breast cancer. However, it has been observed in up to 30-60% of patients with metastatic TNBC. Be careful here- in breast cancer high PD-L1 can be a marker of an IFN-high TME that is associated with a good prognosis.

8) Figure 2I should be data not a cartoon- to show that treatment didn't impact metastasis upon IMFP rechallenge. So these are outgrowth of existing micromets? Or due to the rechallenge? Would be better to use different labels to distinguish them. If you left those mice longer would they have had metastasis progression without rechallenge? How did PD-1/PD-L1 compare between micromets and the recurrent mets? Were there T cells in the microenvironment? Need to show that anti-PD1 alters recruitment (via travel from LN? of previously exhausted cells?) For figure 3- the authors discuss the fact that there may be exhausted T cells in the TME- did they check for this? Earlier pics (supp 3) didn't show a T cell infiltrate in mets at all??

9) Figure 3c and D- the triple combination was not more effective than double or even just chemotherapy in the study 7 day post therapy. This is likely due to the nature of the experiment and timeline- injecting cells and starting treatment 19 days later and then analysing after just 7 days is likely to mainly pick up proliferative/cell death changes. Immune changes may take longer than that. This just needs to be clear in the text.

10) Figure 3E,F, S4E- it is important to show the change in T cells in the TME across multiple lesions and in different mice. This includes for the proximity data- certainly, the addition of immune checkpoint inhibition did not alter the number and proximity of T cells beyond that of double combination treatment based on the data included.

11) Figure 3G- it is not clear why addition of ICB on top of the double treatment would reduce the expression of PD-1/PD-L1? It should be blocking the interaction. How is expression reduced?

12) Figure 4B- the euthanasia on tails of mice treated with triple combination yet without maintenance dose should have been exclusion points on the graph rather than a step down- this would account for the lack of survival benefit.

13) Figure 4E, 4F- it is difficult to determine the spatial co-localisation of CD20 and CD8 positive cells on the images provided- particularly given the focus on one region. Additionally, it is clear that the addition of anti-PD-1 adds nothing in term of the markers analysed in 4F- can a comment to this effect be made in the text?

14) The lung and liver models have all been done using T12 tumor cells- could this be repeated in another model?

15) Figure 5B- is there a positive control tissue to confirm that the CD8 antibody is indeed working on the mouse tissue? Figure 5C,G- can these tissues be stained with a CK antibody to pick up tumour regions more specifically? It is very difficult to see (or quantitate) with the current images.

16) Figure 6D- again there is no significant impact of addition of the maintenance dose of anti-PD-1 on survival (compared to triple combination without maintenance dose). Were the liver mets resected and histology undertaken to confirm FDG-PET results?

Reviewer #3

(Remarks to the Author)

This is an interesting work, exploring in murine models of breast cancers the impact of the modulation of the macrophage population which was found associated with a poor prognosis in TNBC, in particular in a clinical trial exploring a combination of IO and cytotoxic in the same disease. The authors use GEMMs which closely resemble to human models. Their experiments enable to show the value of blocking CSF-1R pathway, in combination with PD1 and CT in lung and liver metastatic models. The models are convincing, and the data generated could prompt the development of a clinical trial exploring a similar Ab and triple combinations in humans. The data presented are convincing. Several questions remain to better assess the relevance of these observations to humans:

1) CTC used is suboptimal from TNBC or equivalent. It would be important to describe in more details in the authors have explore cytotoxics commonly used in humans.

2) Novel CSF1R inhibitors are available including TKI, better tolerated than pexidartinib. could the authors develop this question further, whether Ab and TKIs could provide similar results. What is the specific contribution of the Ab vs a TKI?

3) The models explore different sites of metastases. but in the clinic, often multiple sites are involved with large tumor cell volume. Can the authors expand on the question of metastatic cell load, multiple metastatic sites, could impact on these results (and the duration of efficacy).

The titles of the subchapters should be simplified.

The paragraph of conclusion is not clear and several sentences in the text require review/rewording.

Reviewer #4

(Remarks to the Author)

Version 1:

Reviewer comments:

Reviewer #1

(Remarks to the Author)

The authors have addressed all my comments, and the overall quality of the manuscript has improved. In my opinion, it is now suitable for publication.

Reviewer #2

(Remarks to the Author)

Reviewer #4

(Remarks to the Author)

Reviewer #1 (Remarks to the Author):

In the manuscript by Pedroza et al, the authors developed a new combination treatment for TNBC which is now entering the first phase of a clinical trial. They successfully demonstrated the ability of this treatment to increase survival in a TNBC model. In addition, they tested its effect on lung and liver metastatic models confirming its efficacy.

The manuscript is well written, and it introduces a new treatment for TNBC. However, there are some major issues that need to be addressed and the mechanism behind the therapy efficacy is lacking.

Major comments:

- The rationale for using axatilimab-csfr instead of the frequently used PLX-3397 was to reduce off target effects as PLX-3397 is accompanied with toxicity. However, except for monitoring the tumor weight, in this study there are no analysis related to the toxicity of the anti-CSF1R compound and the combination with CTX. This small inhibitor is reported to induce liver toxicity, so it is important to prove that the mAb does not result in similar adverse effects as the ones induced by PLX-3397.

We thank the reviewer for appreciating the novelty of our study, and for helping to improve the manuscript. As detailed below we have clarified and addressed most of the concerns with both revisions in the text and the inclusion of additional results.

The question of liver toxicity is important and the principal reason that Axatilimab was chosen for these studies. Most importantly, in the clinical trial (NCT04710576, PMID: 39292927) no liver injuries were reported at varies doses "Axatilimab-driven laboratory abnormalities were not accompanied by end-organ damage, a finding consistent with previous reports". Similar results have been mentioned in an unpublished breast cancer trial being performed at Dana Farber. The initial goal of the NEWSTART clinical trial (NCT06959537) will be to assess dose and toxicity of single agents and combination treatments.

In our preclinical studies, we initially tested the mAb alone vs it's respective IgG Control in tumor bearing mice. We proceeded to fix and stain the liver and tumor tissues. We demonstrate that in tumor bearing mice the mAb significantly depletes F4/80+ macrophages in the tumor sparing CD206+ tissue resident macrophages in the liver. These levels were consistent with those of IgG Control treated mice. All the mice were alert and responsive and did not exhibit any unusual behavior or needed to be euthanized. We have added these results as Supplementary Figure 1D and a brief discussion on page 5.

- In the first experiment the authors tested the combination treatment in five different models and observed a strong increase in survival in two of them. In the remaining part of the study, they use mostly only one of these responsive models. It is critical to understand why the other models had only a partial, as these could represent non- or less-responsive patients. In addition, they only show the depletion effect of the anti-csf1r mAb on the T12 model (Fig S1D). Is the therapy not as potent in the other models because the Mf are not well depleted? Or because the niche is filled by other immunosuppressive cells such as neutrophils? Understanding the resistance mechanism and proposing a combination therapy for the unresponsive models would strongly improve the impact of the manuscript.

We do observe similar levels of macrophage depletion in all the models tested following SNDX-ms6352 treatment via IHC as now included as a Supplementary Figure 1J. In two models in which we did not observe a statistically significant response. We have previously shown that the T11 model has a G12V Ras mutation (PMID: 35442423) and is driven by this mutation as it only responds to the Erk1/2 inhibitor Binimetinib (separate manuscript under preparation). The 2208L model is highly infiltrated by S100A8+ neutrophils (Fig. 1A and Supplementary Figure 1C). As reported in Yuan et al (PMID: 39287984) these tumors respond to other inhibitors that target the production of neutrophils. "We have previously reported that when we deplete the macrophages there is a significant increase in neutrophil recruitment (Kim et al, PMID: 31451770), compensating for the loss of those macrophages thus making the tumors less responsive" included in the text on page 16.

- In the first paragraph the authors reported to have treated "several Trp53 null TNBC GEMMs..". First, the authors claim to use GEMM models, however these are transplantable tumors which are not GEMMs. Secondly, "several" is a vague term, they use five models, of which three are claudin low, one is basal like and one luminal like. The text should be more precise and clear for the reader. Especially because the authors claim to have a treatment for TNBC, but it seems that this treatment would be efficient only in a subset of patients with a claudin-low trp53 null tumor.

These transplantable models were all generated from deletion of p53 in Balb/c mice which have been extensively characterized in our laboratories and referred to as GEMMS (PMID: 21633010). To avoid confusion, we now refer to them as "transplantable syngeneic models or syngeneic models".

- The manuscript heavily relies on the two “metastatic models” obtained with tail vein injection for the lungs and portal vein injection for the liver. Although these are easy to obtain models, as the authors highlight in the discussion, these models of experimental metastasis do not reflect the properties of spontaneous metastasis. Hence, some of the results should be recapitulated in a spontaneous metastatic model after resection. This model would reflect better the reality of patient treatment. Moreover, it is not clear why mice with lung metastases could not survive the PET/CT scan, this is frequently done.

To develop combination treatment strategies for mice **with a sufficient metastatic burden** that models stage IV TNBC it is necessary to have sufficient numbers of mice usually 7 or 8 per group in each treatment arm. Please note this is not trivial even for portal vein injection. This is not feasible with any of the existing autochthonous TNBC breast cancer models. Following tumor resections all of our models studied to date specifically the macrophage-enriched tumor models (claudin-low) quickly grew back within the peritoneal cavity and the mice had to be euthanized as required by our animal protocol. Surgical measures including cauterizing blood vessels and fat pad removal after resection were then tried, unfortunately 100% of the mice generated tumors within the abdominal cavity again before any metastases arose. We have reported previously the levels of spontaneous metastases from several of our (non-claudin low) p53 null mouse models (Kim et al, PMID: 31289359) and gene expression analyses demonstrated that the spontaneous and experimental lung metastases were quite similar. We have discussed this previously in PMID: 39084494. Notwithstanding the use of the experimental metastasis models, our study still illuminates the changes within the TIME of lung and liver metastases. Discussed on page 18.

The T12 model was chosen for the experimental metastasis studies because it represents both the TNBC patients with the poorest prognosis (see Singh et al. PMID: 35442423) and represents the high macrophage patient population that will be screened for the NEWSTART clinical trial via pre-treated biopsies.

-The mice that underwent PET/CT scans were bearing established lung macro-metastases and the instrument that we utilized requires these mice to be under anesthesia for a minimum of 1 hour. Following several attempts, we lost approximately 50% of these mice and were advised to not use mice bearing lung-metastasis in fear of losing more due to respiratory arrest.

- A major point is that in general, the mechanism behind the efficiency of the treatment is not clear. The authors make several observations regarding cell infiltration, cytokines levels, but no real mechanism is put forth. Are the remaining Mf repolarized in a way they induce a Th1 response, leading to IFN γ accumulation, subsequently leading to PDL1 upregulation? Is the therapeutic effect fully T-cell mediated? What is the role of B cells and NK cells in the therapeutic efficacy? What is the role of IL-17 and IL-5 and by who are these produced. Is the

cytokine production linked to CSF1R signaling as mentioned in the abstract? At least some mechanistical insights behind the mode of action should be provided.

To provide a mechanistic insight we have now added an additional supplementary figure (Figure S3) to our manuscript. Here we demonstrate that the combination specifically increases CD4⁺/CD8⁺ T cells and B cell infiltration (Figure S3A and B) in primary tumors. Further, we also confirm increased CCL signaling between neutrophils and monocytes/macrophages which leads to the observed increase in mature macrophage populations following CTX alone (Figure S3C). This is accompanied by increased CCL signaling between monocytes and macrophages directly with NK cells which could lead to their suppression as previously observed. Interestingly, while CTX upregulates both PD-L1 (CD274) and MCH II (H2-Ab1) levels within the various macrophage populations (Figure S3D and E), combination treated tumors decrease PD-L1, with residual monocytes and inflammatory-TAMs remaining MHC II positive. "Mechanistically the residual inflamed-TAM (MHC II⁺) and previously observed Fabp5⁺ LA-TAMs may coordinate a long-term anti-tumor response by attracting CD4⁺/CD8⁺ T and B cells. Within the combination treated tumors, CD4 T and dendritic cell increased interferon gamma receptor 1, the specific receptor for IFN- γ , while SNDX-ms6352 alone downregulated *CSF1R* in both monocytes and macrophages (Figure S3E). These results were confirmed by IHC whereby SNDX-ms6352 efficiently depleted F4/80, CSF-1R and the immunosuppressive macrophage marker ARG1 in primary tumors (Figure S3F)" has been added to the Results section along with the figure and figure legend on page 16.

- Finally, is the addition of anti-PD1 treatment really necessary? In Fig. 2C we can observe that approximately 70% of the mice treated with the combination are alive at 56 days. In fig. 4B mice treated with the same combination + anti-PD1 are all dead at this time point. Can the authors clarify on this? In addition, the mice receiving the m. dose of PD1 were only monitored until 70 days. Did the surviving mice showed sign of metastasis? Also in the case of liver metastasis, Fig. 6D, the mice were only monitored until day 56, these mice could still developed metastasis. After the rechallenge, did the authors check the livers of these mice?

In Fig 4B we included an extra group to directly compare CTX+anti-PD1 without SNDX-ms6352. All these mice died by 50 days and there were no long-term survivors. This helps support the conclusion that macrophage depletion is necessary to prolong survival.

-Because in our previous experiment (Figure 2) we monitored the mice for 56 days we proceeded to extend this for an extra 2 weeks to assess recurrence. The mice that survived up to 70 days that were treated with the anti-PD1 maintenance(m.) dose were rechallenged and none of them exhibited any lung metastases unlike our previous experiment where 100% of the mice that were treated with the double combination (Figure 2) exhibited lung metastasis recurrence following rechallenge.

-We did not observe any liver metastases following the liver rechallenge and thus we deemed these mice "cured" as depicted in our schematic (Fig 6I), prior to performing the adoptive T cell transfer from the same mice" discussed on page 13.

Minor comments:

In general the figures are quite blurry and difficult to read. This might be due to the compression during the submission, but it would be good to have high quality figures in the revised version.

-Since we used high dimensional IMC and Immunofluorescence the size of the figures that could be uploaded was capped so the figures were compressed. For publication we will submit power point slides with high resolution images, and we will happily provide them for the reviewer if necessary.

- Fig 1F and S1J: the authors claim that these two models are resistant to tumor rechallenge. However, we can still observe tumor growth, which appears to be just delayed, so the models can't be really considered "resistant". The authors should rephrase their claims.

-We agree with the reviewer and have changed our language to "delayed tumor cell growth following contralateral mammary tumor cell re-challenge" page 6. to omit claiming that they are "resistant".

- Some typos, such as in the abstract: IL-15 instead of IL-5, Several times INF is written instead of IFN

-We have corrected these typos within the text, including IL-5 and IFN.

- Fig 1J, how was this circle plot analysis performed?

-“The circle plot was generated by CellChat (netVisual_circle function)”. We now have described the specific bioinformatic guidelines for this analysis in the Materials and Methods section, p. 28.

- Fig 1G, in the umap some names are covering each-other

-We have reformatted the UMAP names and included a key (Fig1G).

- Fig 1K, the plasma from IgG control mice is missing from this analysis.

-All the levels are normalized to IgG controls depicted by the line. We have now included this in Figure legend 1K, page 40 and under Materials and Methods, page 24.

- Fig. 2F and S3D: what are con1&2 and Combos 1&2? Were only 2 lungs used for the analysis? Fig. S3E-F: what is considered a MaMet and what a MiMet? In general, for all the IMC experiments is not clear how many samples were analyzed.

We performed IMC on two independent lungs for the analysis but measured n=3 ROIs per lung and now have included this in the figure legend, page 61. MaMet are Macro-metastases and MiMET are micro-metastases, now included in the figure legend, page 61.

- What is the difference between 2B and S2A?

-These are representative images of independent lungs bearing established metastases at the same time point as now included in the figure legend, page 61.

- Page 10, referring to fig. 4F the authors talk about double combination, while the staining has been done on the triple combination.

-We stained for single agent, double and triple combinations depicted by Fig 4F and IF images included as sup Fig 5A.

- Fig. 6C, in the legend is reported that the analysis has been done on 3 mice per group, but on the figure there are represented only 2 dots.

We depicted 3 dots but two of the controls are overlapping each other, the diamonds and the lines show 3 independent replicates. To clarify, we now plotted one of the lines as a dotted line (Fig 6C) and have included the raw data in the source data files.

Reviewer #2 (Remarks to the Author):

This group builds on previous data published in Cancer Research (PMID: 35442423) to investigate the targeted inhibition of CSF-1R in combination with cyclophosphamide (CTX), along with anti-PD1 in experimental lung and liver metastasis using their transplantable GEM models that recapitulate mutant-p53 claudin-low subtypes. The previous paper did include analysis of the dual combination on orthotopic tumor growth (in a couple of GEM-derived models)– yet this paper brings in metastasis and addition of anti-PD1. Targeting CSF-R1 for breast cancer metastasis, alone or in combination with chemotherapy or PD-1 antibodies, is not novel, However, the use of the more specific SNDX-ms6352 antibody and the mechanisms at play in the TME are important to guide future clinical trials.

Main points that reduce enthusiasm for publication is the lack of metastasis models used, which is important when using experimental rather than spontaneous models. The authors focus the metastasis studies on one model- the T12 model. Given that the addition of anti-PD1 had most benefit in the liver metastasis model (more-so than lung), this should be expanded to determine the clinical relevance for publication in Nature Comms.

Specific comments/questions are:

1) Figure 1D- only the highly neutrophil enriched luminal-like 2208L model failed to respond to the combination. Given this model still has more macrophages than neutrophils, what is going on here? Need to block both? It would have been interesting to include this model in the triple combination to enable the link between response and the state of the TME pre- and post-therapy.

As depicted in Figure 1 the 2208L model is neutrophil enriched and lacking in abundant macrophages compared to the claudin-low model. We recently have published a separate study targeting the production of neutrophils using a unique p300/CBP bromodomain inhibitor (Yuan et al PMID: 39287984). As highlighted in the Introduction and Discussion the current preclinical study is focused on the treatment of EMT and macrophage enriched TNBC which nicely phenocopy stage IV metastatic TNBC patients, which have the poorest prognosis. We have included a paragraph describing this on page 16.

2) Figure 1G- an impressive difference in T an B cells and decrease in macrophages with combination- but was this the only replicate for each treatment?

This was from 3 independently treated mouse tumors, n=3 mice/treatment (updated in the Figure legend) page 40.

3) Figure 1J- is this really the best way to depict this? It's confusing.

The circle plots allows us to demonstrate clear communication disruption by the thinning of lines between tumor cells and other immune cells that rely on CSF signaling. This is one approach to illustrate the scRNA-seq data.

4) Figure 1K- as far as I can see, the no treatment control is missing here and that should be included to determine if any single agent is detrimental or the opposite trend is observed.

All the levels are normalized to IgG controls depicted by the lines, and we have now included this in the figure legend, page 40 and in the methods section page 24.

5) Figure 2C- "only the combination led to better survival even after treatment cessation" this is not correct as chemotherapy alone did as well. This is critical given the novel angle of this paper and that, in figure 4, the addition of anti-PD1 with CTX is no different to the triple combination.

We agree and for clarity now have stated in the Results, page 8 that "only single agent chemotherapy or the combination increased the survival outcome compared to IgG control and SNDX-ms6352. However, compared to chemotherapy alone the combination does significantly prolong overall survival".

6) Figure 2F/G- Although Combos 1&2 expressed decreased proliferative tumor cells (Ki67+, Vim+, CD44+) these immunologically cold micro-metastases displayed elevated levels of exhaustion markers PD-L1 and PD-1- from the data presented this is incorrect and only applies to PD-1 as the tumour cells in the control already had high PD-L1. Why was only combination of controls? Single agents should be looked at as well.

While one of the controls did have increased PD-L1 levels, we cannot infer that all the tumors initially had high PD-L1 levels. As shown in Supplementary Fig 4, Control 1 has almost undetectable levels of PD-L1 and Control 2 has significantly lower levels when compared to both combo treated as shown in Fig 2F. It's not until we treat them that we observed significant increased levels of PD-L1 which was confirmed in a separate independent experiment shown in Fig 3E where increased PD-L1 levels were only seen following the double combination. In this case the controls also have undetectable PD-L1. To clarify single agents were analyzed as shown in Fig 3.

7) Page 9- PD-L1 is a well-known T cell exhaustion marker that is not commonly observed in non- metastatic breast cancer. However, it has been observed in up to 30-60% of patients with metastatic TNBC. Be careful here- in breast cancer high PD-L1 can be a marker of an IFN-high TME that is associated with a good prognosis.

We concur and have included the following in the Results, page 9, “Elevated levels of PD-L1 can also be a marker of an IFN-high TIME which can be associated with a favorable prognosis” (PMID: 35053471, PMID: 33193888, PMID: 28430626).

8) Figure 2I should be data not a cartoon- to show that treatment didn't impact metastasis upon IMFP rechallenge. So these are outgrowth of existing micromets? Or due to the rechallenge? Would be better to use different labels to distinguish them. If you left those mice longer would they have had metastasis progression without rechallenge? How did PD-1/PD-L1 compare between micromets and the recurrent mets? Were there T cells in the microenvironment? Need to show that anti-PD1 alters recruitment (via travel from LN? of previously exhausted cells?) For figure 3- the authors discuss the fact that there may be exhausted T cells in the TME- did they check for this? Earlier pics (supp 3) didn't show a T cell infiltrate in mets at all??

To address this concern, we now have included the actual H&E and IHC stained images from the lungs of the mice that exhibited metastatic recurrence (Figure 2J) page 41. We agree that it is not possible to determine if these are from existing outgrowths or due to the rechallenge. However, we suspect this could have been from existing micro-mets that presented with ITH based on our Sup Fig 3F/G and were reawakened following the tumor cell rechallenge. This is why we followed the mice up to 70 days as detailed in the Results on page 45 (Fig4B) to assess for recurrence. There were no T cells present in the recurrent lung mets Figure 2J page 41. Based upon our analysis we observed varying levels of CD8/CD4 specific T cells following the treatments (Sup Fig5D) and accordingly we now have clarified this as follows, “we speculate that the high levels of PD-L1 could lead to T-cell exhaustion if T cells begin to infiltrate” in the Results on page 9.

9) Figure 3c and D- the triple combination was not more effective than double or even just chemotherapy in the study 7 day post therapy. This is likely due to the nature of the experiment and timeline- injecting cells and starting treatment 19 days later and then analysing after just 7 days is likely to mainly pick up proliferative/cell death changes. Immune changes may take longer than that. This just needs to be clear in the text.

One intrinsic limitation of this experimental design is due to the nature of the single agent controls which must be euthanized quickly. Therefore, we needed to collect tissue at the same timepoint to compare the TME across all treatments. As the reviewer points out double and triple combo had similar met scores (Fig3C), we now have explained this in the Results, pages 9 and 10.

10) Figure 3E, F, S4E- it is important to show the change in T cells in the TME across multiple lesions and in different mice. This includes for the proximity data- certainly, the addition of immune checkpoint inhibition did not alter the number and proximity of T cells beyond that of double combination treatment based on the data included.

We now have included additional single and combination images from our IMC analysis (Sup Fig4E) page 62. The distance analysis was conducted from multiple ROIs and calculated from single cells following cell segmentation (Single cell raw files were uploaded as Excel Sup Files) to capture maximum number of cells.

11) Figure 3G- it is not clear why addition of ICB on top of the double treatment would reduce the expression of PD-1/PD-L1? It should be blocking the interaction. How is expression reduced?

In breast cancer models, ICB combined with radiation therapy has been shown to decrease PD-L1 levels on both tumor and immune cells (PMID: 24382348). In patients with melanoma pembrolizumab lowered PD-L1 expression in post-treatment biopsies due to tumor regression (PMID: 25428505), the authors point out that although PD-L1 expression is often induced by IFN- γ , as the tumors regress after therapy, the signal diminishes.

The reviewer pointed this out in a previous comment and as such we have now included our explanation, “PD-L1 is often upregulated in response to stress or inflammation, however we speculate that the levels of PD-L1 may drop as the metastatic burden diminishes in response to a shift within the TIME leading to less PD-L1+ Vim+CD44+ tumor cells” (PMID: 24382348 and 25428505). These outcomes are consistent with dynamic regulation of PD-1/PD-L1 in response to changes in tumor burden and immune tone. Please see page 10.

12) Figure 4B- the euthanasia on tails of mice treated with triple combination yet without maintenance dose should have been exclusion points on the graph rather than a step down- this would account for the lack of survival benefit.

We agree and we now have excluded them and noted this in the Results and figure legend. Pag 45.

13) Figure 4E, 4F- it is difficult to determine the spatial co-localisation of CD20 and CD8 positive cells on the images provided- particularly given the focus on one region. Additionally, it is clear that the addition of anti-PD-1 adds nothing in term of the markers analysed in 4F- can a comment to this effect be made in the text?

For this study we were surprised to see these types of residual structures that we had not observed before, even in our primary tumors. We have accordingly added the following: “Although we did detect increased levels of CD20 and CD8 in the double and triple combination, these residual structures were only observed in the triple combination within the lung”, page 11. As we noted in the Results we suspect that these could be possible TLSs and may be a good indicator for a favorable prognosis, this was just one representative image of multiple structures from multiple mice (n=5 independently treated mice).

14) The lung and liver models have all been done using T12 tumor cells- could this be repeated in another model?

This study was designed to treat established metastases of the tumor model (T12) because it best represents the high macrophage TNBC patient population that has been demonstrated to have a worse pathological complete response to neoadjuvant chemotherapy, (PMID: 34253579). Efforts to profile the primary and metastatic TIME of several models to understand why some models generate experimental metastasis in lung and or liver and some don't and their differential treatment responses to different combination therapies are ongoing with our co-author and collaborator Dr. Charles Perou. These studies are beyond the scope of this manuscript.

Most importantly, however pre-treated biopsies from TNBC patients will be screened and only patients that exhibit the high macrophage/EMT criteria will be enrolled into the NEWSTART clinical trial (NCT06959537). The NEWSTART clinical trial will provide the appropriate validation of this model.

15) Figure 5B- is there a positive control tissue to confirm that the CD8 antibody is indeed working on the mouse tissue?

Figure 5C,G- can these tissues be stained with a CK antibody to pick up tumour regions more specifically? It is very difficult to see (or quantitate) with the current images.

The same CD8 antibody was utilized in multiple other studies (PMID: 37874652, PMID: 39287984, PMID: 39419025) and throughout our experimental treatments. Although we did not detect any T cells in the liver initially which we deemed to be immunologically “cold” the same antibody was highly detected (CD8) following the triple combination in the liver and lung (Sup Fig 7B and Sup Fig 5B). Furthermore, the same antibody was also used to detect the T cells that were transferred into tumor bearing mice, both in the tumor and the spleen tissues (Sup Fig 7 F and G).

We agree with the reviewer that these are difficult to be observed, thus we outlined the metastatic lesions with a dotted line. Cytokeratin antibody staining was not specific for these mesenchymal tumors. Page 47

16) Figure 6D- again there is no significant impact of addition of the maintenance dose of anti-PD-1 on survival (compared to triple combination without maintenance dose). Were the liver mets resected and histology undertaken to confirm FDG-PET results?

Although statistically the impact was not as significant, 75% of the mice died when the m.dose was not given compared to only 33%. Furthermore, all of the mice that survived following treatment cessation from the m.dose rejected the tumor cell rechallenge. ACT studies confirmed durable long-term anti-tumor adaptive immunity. This is now discussed on page 12 and 13.

The livers were harvested, stained and had no detectable metastases via H&E as clarified on page 13. We can add these figures if necessary, as supplemental.

Reviewer #3 (Remarks to the Author):

This is an interesting work, exploring in murine models of breast cancers the impact of the modulation of the macrophage population which was found associated with a poor prognosis in TNBC, in particular in a clinical trial exploring a combination of IO and cytotoxic in the same disease. The authors use GEMMs which closely resemble to human models. Their experiments enable to show the value of blocking CSF-1R pathway, in combination with PD1 and CT in lung and liver metastatic models. The models are convincing, and the data generated could prompt the development of a clinical trial exploring a similar Ab and triple combinations in humans. The data presented are convincing. Several questions remain to better assess the relevance of these observations to humans:

1) CTC used is suboptimal from TNBC or equivalent. It would be important to describe in more details if the authors have explored cytotoxics commonly used in humans.

We thank the reviewer for appreciating the impact of our study, and for helping to improve the manuscript. As detailed below we have clarified and addressed most of the concerns with revisions in the text.

We have added the following “Low-dose cyclophosphamide was selected specifically for its immunomodulatory effects—particularly its documented ability to selectively deplete regulatory T cells (Tregs), thereby potentially enhancing anti-tumor immunity in combination with immunotherapy agents. This approach has been previously established and extensively described in the literature; wherein low-dose cyclophosphamide has demonstrated efficacy in modulating the immune microenvironment, promoting T-cell-mediated responses, and reducing immunosuppressive barriers within the tumor milieu PMID: 28855352, PMID: 36097618, PMID: 15170446” on page 15. Our published preclinical studies document the importance of including cyclophosphamide as part of this unique combination therapy (PMID: 35442423).

2) Novel CSF1R inhibitors are available including TKI, better tolerated than pexidartinib. could the authors develop this question further, whether Ab and TKIs could provide similar results. What is the specific contribution of the Ab vs a TKI?

We have elaborated on the off targets of TKIs such as pexidartinib that differentiate them from Abs which are more specific. “While pexidartinib is a CSF-1R inhibitor, it also exhibits activity against c-Kit and FLT3 PMID: 28716061. These off-target effects may be the cause for the liver toxicity that resulted in PLX-3397 being discontinued in the clinic for the treatment of TNBCs. The lack of liver toxicity of other small molecule TKIs has not been demonstrated in patients, while the lack of toxicity of Axatilimab has, and Niktimvo is FDA approved” on page 15.

3) The models explore different sites of metastases. but in the clinic , often multiple sites are involved with large tumor cell volume. Can the authors expand on the question of metastatic cell load, multiple metastatic sites, could impact on these results (and dthe duration of efficacy).

We agree with the reviewer that metastatic burden and the presence of multiple involved organs are key determinants of therapeutic response and clinical durability. We have added the following to the discussion “In our models, we studied single-organ metastases to isolate organ-specific effects and treatment mechanisms. However, we acknowledge that in patients, metastases can be multi-site and associated with higher tumor load. This may influence the TIME, potentially altering therapeutic efficacy” on page 18.

The titles of the subchapters should be simplified.

We have simplified all of the subheadings.

The paragraph of conclusion is not clear and several sentences in the text require review/rewording.

We have carefully edited the conclusion to address the reviewers’ comments and edited the text for clarity.

Reviewer #4 (Remarks to the Author):
